# Structural basis for +1 ribosomal frameshifting during EF-G-catalyzed translocation

Gabriel Demo[1,2], Howard B. Gamper [3], Anna B. Loveland [1], Isao Masuda [3], Christine E. Carbone[1], Egor Svidritskiy[1], Ya-Ming Hou [3✉] & Andrei A. Korostelev [1✉]

Frameshifting of mRNA during translation provides a strategy to expand the coding repertoire of cells and viruses. How and where in the elongation cycle +1-frameshifting occurs remains poorly understood. We describe seven ~3.5-Å-resolution cryo-EM structures of 70S ribosome complexes, allowing visualization of elongation and translocation by the GTPase elongation factor G (EF-G). Four structures with a + 1-frameshifting-prone mRNA reveal that frameshifting takes place during translocation of tRNA and mRNA. Prior to EF-G binding, the pre-translocation complex features an in-frame tRNA-mRNA pairing in the A site. In the partially translocated structure with EF-G•GDPCP, the tRNA shifts to the +1-frame near the P site, rendering the freed mRNA base to bulge between the P and E sites and to stack on the 16S rRNA nucleotide G926. The ribosome remains frameshifted in the nearly post-translocation state. Our findings demonstrate that the ribosome and EF-G cooperate to induce +1 frameshifting during tRNA-mRNA translocation.

[1] RNA Therapeutics Institute, Department of Biochemistry and Molecular Pharmacology, UMass Medical School, Worcester, MA, USA. [2] Central European Institute of Technology, Masaryk University, Brno, Czech Republic. [3] Department of Biochemistry and Molecular Biology, Thomas Jefferson University, Philadelphia, PA, USA. ✉email: Ya-Ming.Hou@jefferson.edu; Andrei.Korostelev@umassmed.edu

To accurately synthesize a protein, the ribosome maintains the mRNA reading frame by decoding and translocating one triplet codon at a time. Concurrent ~25 Å movement of the mRNA and tRNAs is catalyzed by the conserved translational GTPase EF-G in bacteria (EF2 in archaea and eukaryotes). After formation of a peptide bond, the peptidyl-tRNA, and deacylated tRNA move from the A and P sites to the P and E sites, respectively. This translocation requires spontaneous and large-scale (~10°) inter-subunit rotation of the ribosome[1,2]. Despite pronounced rearrangements of subunits and extensive motions of tRNA and mRNA at each elongation cycle, the ribosome maintains the correct reading frame through hundreds of codons[3].

Nevertheless, change of the reading frame, termed frameshifting, is common in viruses, bacteria, and eukaryotes, where it enables expansion of the coding repertoire and regulation of gene expression[4]. During frameshifting, the translating ribosome switches to an alternative reading frame, either in the forward (+) or reverse (−) direction, i.e., skipping or re-reading one or more mRNA nucleotides, respectively. This work focuses on +1 frameshifting (+1FS), which is important for gene expression in various organisms. For example, +1FS controls the expression of the essential release factor 2 in bacteria[5,6], regulates metabolite-dependent enzyme expression[7], and leads to pathological expression of huntingtin[8] in eukaryotes. +1FS can be amplified by dysregulation of ribosome quality control mechanisms[9,10], and it is being exploited to synthetically expand the coding repertoire of genomes by inserting non-natural amino acids via a tRNA that can perform +1FS[11]. Because +1FS occurs during the dynamic stage of protein elongation, its molecular mechanism has been challenging to study.

Here we address this challenge by using cryo-EM (CE) to visualize +1FS on one of the most +1FS-prone ("slippery") mRNA sequences in bacterial genomes. The mRNA sequences CC[C/U]-[C/U][12] induce +1FS due to imbalances in tRNA concentrations[13,14], lack of tRNA post-transcriptional modifications[15–18], or nucleotide insertions to the anticodon loop of tRNAs[18–21]. Under normal growth conditions, CC[C/U]-[C/U] sequences in *Escherichia coli* can induce +1FS up to ~1%[22], exceeding the average frequency of spontaneous frameshifting at other sequences by two orders of magnitude[23]. In vitro, mRNA CC[C/U]-N (N = A, C, G, U) sequences are even more prone to +1FS, achieving 70% efficiency[17]. The CC[C/U]-N sequences code for proline (Pro) and are decoded by two isoacceptors of tRNA$^{Pro}$ in *E. coli*[17]. The isoacceptor tRNA$^{Pro}$ (UGG), encoded by *ProM*, is essential for cell growth due to its ability to read all four Pro codons[24]. It is highly prone to +1FS upon loss of the post-transcriptionally modified nucleotides 5-oxyacetyl uridine 34 (cmo⁵U34)[17] and/or $N^1$-methylation of guanosine 37 (m$^1$G37)[22]. The isoacceptor tRNA$^{Pro}$(GGG), encoded by *ProL*, is cognate to the CC[C/U] codon of the slippery motif, where +1FS can be induced upon loss of m$^1$G37 and/or of the elongation factor EF-P[22].

Studies have proposed that +1FS by tRNA$^{Pro}$(UGG) can occur during one of the three stages of the elongation cycle: (1) decoding of a slippery sequence when the tRNA binds to the ribosomal A site[25–27]; (2) EF-G-catalyzed translocation of the tRNA from an in-frame position at the A site to the +1-frame position in the P site;[22] or (3) stalling of the tRNA in the P site after translocation and/or EF-G dissociation[22,28–30]. Crystal structures of anticodon stem-loops (ASLs) of other +1FS-prone tRNAs in the A site[26,31–33], formed in the absence of elongation factors, argue against the shift during decoding, showing that steric hindrance in the decoding center prevents tRNA from slippage. Yet, the dynamics of the ribosome allow sampling of different structures, which may evade crystallization. Thus, the possibility of rearrangements of a frameshifting complex at all

three elongation stages remain to be explored. To distinguish among the above three mechanisms, it is necessary to capture 70S translocation complexes that are formed with full-length aminoacyl-tRNAs and EF-G on a +1FS-prone mRNA.

In this work, we present cryo-EM structures of 70S complexes with full-length native *E. coli* tRNA$^{Pro}$(UGG), formed with and without EF-G. The structures reveal differences between complexes with a non-frameshifting "control" mRNA and those containing a +1FS-prone mRNA. Unlike the ASLs in previous studies of +1FS[26,31–33], which contained an extra nucleotide next to the anticodon[34], native *E. coli* tRNA$^{Pro}$(UGG) has a canonical anticodon loop. Here, we first describe two pre-translocation 70S complexes, containing a non-frameshifting mRNA codon motif $C^1CA$-$A^4$, or the frameshifting codon motif $C^1CC$-$A^4$, in the A site. Each complex is prepared with fMet-tRNA$^{fMet}$ in the P site and Pro-tRNA$^{Pro}$(UGG) delivered by EF-Tu•GTP to the A site. To capture EF-G-catalyzed translocation states, we then add EF-G with non-hydrolyzable GTP analog GDPCP (5′-guanosyl-β,γ-methylene-triphosphate) to each pre-translocation complex and perform single-particle cryo-EM analyses (Methods). We use maximum-likelihood classification of cryo-EM data, which allows the separation of numerous functional and conformational states within a single sample. Our data classification reveals three elongation states in each complex (Supplementary Figs. 1, 2, and 4): (1) pre-translocation non-rotated 70S structures with tRNA$^{Pro}$ in the A site (I: non-frameshifting, and I-FS: frameshifting); (2) "mid-translocation" EF-G-bound structure, with tRNA$^{Pro}$ near the P site (II and II-FS); and (3) nearly fully translocated EF-G-bound state with tRNA$^{Pro}$ in the P site (III and III-FS). In addition, to visualize the pre-translocation ribosome before interaction with EF-G, we analyze the frameshifting complex formed without EF-G, which yielded a rotated pre-translocation 70S ribosome containing tRNA$^{Pro}$ in a hybrid A/P* conformation (I$^{rot}$-FS) (Supplementary Figs. 3 and 4). Comparison of the non-frameshifting and frameshifting structures reveals that the ribosome is pre-disposed for +1FS before translocation, and that frameshifting is accomplished at an intermediate stage of EF-G-catalyzed translocation.

## Results and discussion

**E. coli tRNA$^{Pro}$(UGG) has a propensity for +1FS in vivo and in vitro.** While we previously showed that *E. coli* tRNA$^{Pro}$(UGG) performs +1FS in vitro[17], it remained unknown whether the tRNA performs +1FS in cells, where all isoacceptors are present, and whether its post-transcriptional modifications impact +1FS. We addressed these questions in a cell-based reporter assay, in which a $C^1CX$-$X^4$ codon motif was inserted at the 2nd codon position of the *lacZ* gene next to the start codon AUG (Fig. 1a), such that a +1FS event was necessary to synthesize the full-length β-galactosidase (β-gal). The frequency of +1FS was measured as the ratio of β-gal in cells expressing the CCX-X reporter over cells expressing an in-frame insertion of the CCC or CCA codon to the *lacZ* reporter[22]. We generated a non-FS $C^1CA$-$A^4$ reporter, where the CCA codon was cognate to tRNA$^{Pro}$(UGG), and a +1FS CCC-A reporter, where the $C^2CA^4$ codon in the +1-frame would be cognate to tRNA$^{Pro}$(UGG). We also generated a separate +1FS CCC-C reporter, where the CCC codon was cognate to tRNA$^{Pro}$ (GGG).

In cells that expressed the CCA-A reporter and contained both tRNA$^{Pro}$(UGG) and tRNA$^{Pro}$(GGG), +1FS was suppressed to a background level (0.6%), which remained stable even upon deletion of tRNA$^{Pro}$(GGG) (0.7%) (Fig. 1b). This indicates that tRNA$^{Pro}$(UGG) is sufficient to decode the CCA codon of the reporter in cells. The loss of m$^1$G37 on tRNA$^{Pro}$(UGG) resulted in an elevated +1FS (1.7%) (Fig. 1b), supporting the notion that

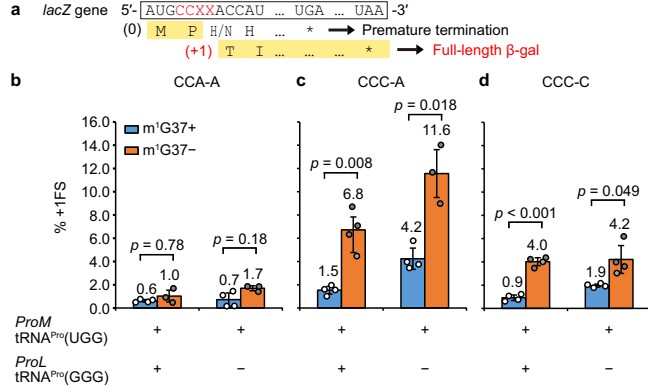

**Fig. 1 +1FS of *E. coli* tRNA^Pro(UGG) in a cell-based assay. a** The *lacZ* reporter construct. A CCX-X codon motif was inserted to the *lacZ* gene next to the start codon AUG, requiring a +1FS event at the motif to produce a full-length β-gal protein. **b–d** Measured +1FS frequency in *E. coli trmD*-KO/JM101 strains expressing the Ara-controlled human *trm5* to produce the m^1G37+ (blue) or m^1G37– (orange) condition. The +1FS frequency was reported for the strain expressing the CCA-A reporter (**b**), the CCC-A reporter (**c**), and the CCC-C reporter (**d**), by determining the β-gal activity as Miller Units normalized by that of the corresponding control strain. Data are shown as mean ± SD, $n = 4$, for all but three datasets. For datasets CCA-A/proM+/proL+/m^1G37–, CCA-A/proM+/proL–/m^1G37–, and CCC-A/proM+/proL–/m^1G37–, $n = 3$. $P$ values are calculated from Welch's two-sided $t$ test and are shown in (**b–d**).

this tRNA isoacceptor is dependent on m^1G37 to maintain the reading frame[17]. By contrast, in cells that expressed the CCC-A reporter (Fig. 1c), expression of tRNA^Pro(UGG) in the absence of tRNA^Pro(GGG) resulted in a substantial increase in +1FS (4.2%) relative to those expressing both tRNAs (1.5%), and the +1FS frequency was further elevated by the loss of m^1G37 (11.6%). These results are consistent with the notion that, while the CCC codon can be read by tRNA^Pro(UGG) through the cmo^5U34-C3 wobble pairing, this pairing is unstable and is prone to +1FS to form a stable cmo^5U34-A4 pairing. In cells that expressed the CCC-C reporter (Fig. 1d), loss of the cognate tRNA^Pro(GGG) increased +1FS from 0.9% to 2.0%, indicating the ability of tRNA^Pro(UGG) to read the 0-frame or +1-frame through cmo^5U34-C3 or cmo^5U34-C4 pairing. The further increase of +1FS upon loss of m^1G37 (to 4.0% or 4.2%) confirmed the importance of this methylation in suppressing +1FS of tRNA^Pro (UGG).

We next used an in vitro translation assay to measure the efficiency of +1FS under different buffer conditions, varying the concentrations of Mg^2+ and other constituents to affect the fidelity and efficiency of translation[35,36]. We compared a high-fidelity (HF) buffer, which contains MgCl₂ at the near-physiological 3.5 mM, and a cryo-EM buffer, which contains MgCl₂ at 20 mM in the presence of spermine and spermidine as stabilizing reagents commonly used to capture ribosome complexes (see "Methods"). We began by testing frameshifting of the native-state tRNA^Pro(UGG) with natural post-transcriptional modifications. We formed an *E. coli* 70S initiation complex (70SIC) with fMet-tRNA^fMet in the P site and an mRNA containing the non-FS sequence AUG-CCA-AGU-U or the +1FS sequence AUG-CCC-AGU-U. We next mixed the 70SIC with an equimolar mixture of ternary complexes of EF-Tu•GTP with Pro-tRNA^Pro(UGG), Ser-tRNA^Ser, and Val-tRNA^Val in the presence of EF-G and GTP (Fig. 2a). Upon translocation and subsequent decoding, the resulting fMPS tripeptide would report on the amount of the 0-frame product, whereas fMPV would report on the +1FS product. As expected, the fMPV tripeptide was

synthesized on the +1FS-prone CCC-A reporter but not on the CCA-A reporter (Fig. 2b–d). Measurement of the +1FS frequency, based on the fractional conversion of fMP to fMPV and fMPS, showed that the +1FS frequency was higher at 37 °C than at 20 °C (Fig. 2b, c), in keeping with the dependence of +1FS on the dynamics of the ribosome. The +1FS frequency was generally higher in the HF buffer than in the cryo-EM buffer, consistent with the notion that high Mg^2+ concentrations are inhibitory to protein synthesis[35,36]. Indeed, translation of the 0-frame fMPS decreased with increasing concentrations of Mg^2+ in both the HF and cryo-EM buffers (Supplementary Fig. 5).

We next measured how post-transcriptional modifications in tRNA^Pro(UGG) contribute to +1FS by performing the assay with the fully modified native-state tRNA^Pro(UGG) or a modification-free in vitro transcript of tRNA^Pro(UGG). The unmodified tRNA resulted in increased +1FS on the CCC-A motif, but not on the CCA-A motif (compare Fig. 2e, f with 2c, d). This indicates that the absence of the cmo^5 modification on U34 (the only post-transcriptionally modified nucleotide in the anticodon) likely destabilizes the wobble U34-C3 base pair, and that this destabilization increases +1FS on the CCC-A motif. Notably, the transcript tRNA was slower in the fractional synthesis of the 0-frame fMPS at the CCC-A motif and failed to reach a plateau (Supplementary Fig. 6a, b), indicating inefficient interaction of the unmodified tRNA with the ribosome. These experiments demonstrate that our purified native-state tRNA^Pro(UGG) is an efficient substrate for protein synthesis and that it is capable of inducing +1FS both in cells and in functional assays in vitro.

**Pre-translocation frameshifting structures adopt an open 30S conformation.** Decoding of mRNA occurs on the non-rotated ribosome, in which peptidyl-tRNA occupies the P site and an aminoacyl-tRNA is delivered by EF-Tu to the A site. Universally conserved 16S ribosomal RNA nucleotides of the decoding center G530, A1492, and A1493 (*E. coli* numbering) interact with the codon–anticodon helix, resulting in the closure of the 30S domain[37], which stabilizes the cognate aminoacyl-tRNA during decoding[38]. Peptidyl transfer results in a deacylated tRNA in the P site and the peptidyl-tRNA in the A site, preparing the ribosome for translocation[3]. Thus, the closure of the 30S domain is a signature of canonical decoding at the A site.

We performed cryo-EM analyses of ribosomes with non-frameshifting CCA-A or frameshifting CCC-A motifs, and first focused on the non-rotated pre-translocation structures (Fig. 3; "Methods"). cryo-EM data classification reveals differences between the non-frameshifting and frameshifting complexes formed with EF-G•GDPCP (Fig. 3). While particle populations are similar (~11% and ~12%, respectively), consistent with comparable efficiency of decoding of both mRNA sequences[17], the resulting CE maps report different conformations of the 30S subunit. The non-frameshifting Structure I features a canonical "closed" 30S subunit, in which G530, A1492, and A1493 are in the ON state[38] and interact with the backbone of the cognate codon–anticodon helix (Fig. 3b, c). G530 contacts A1492, resulting in a latched decoding center nearly identical to that in cognate 70S complexes formed with other tRNAs[39,40]. By contrast, the frameshifting Structure I-FS with the cmo^5U34-C3 wobble pair features an open 30S conformation (Fig. 3e, f), in which the 30S shoulder domain is shifted away from the 30S body domain. This open conformation resembles transient intermediates of decoding captured by cryo-EM and is preferred when mismatches are present in the codon–anticodon duplex[38,41]. Here, G530 (at the shoulder) is retracted by ~2 Å from the ON position, shifting away from A1492 (at the body) and from the backbone of G35 of tRNA^Pro (Fig. 3f). Thus, the decoding-center

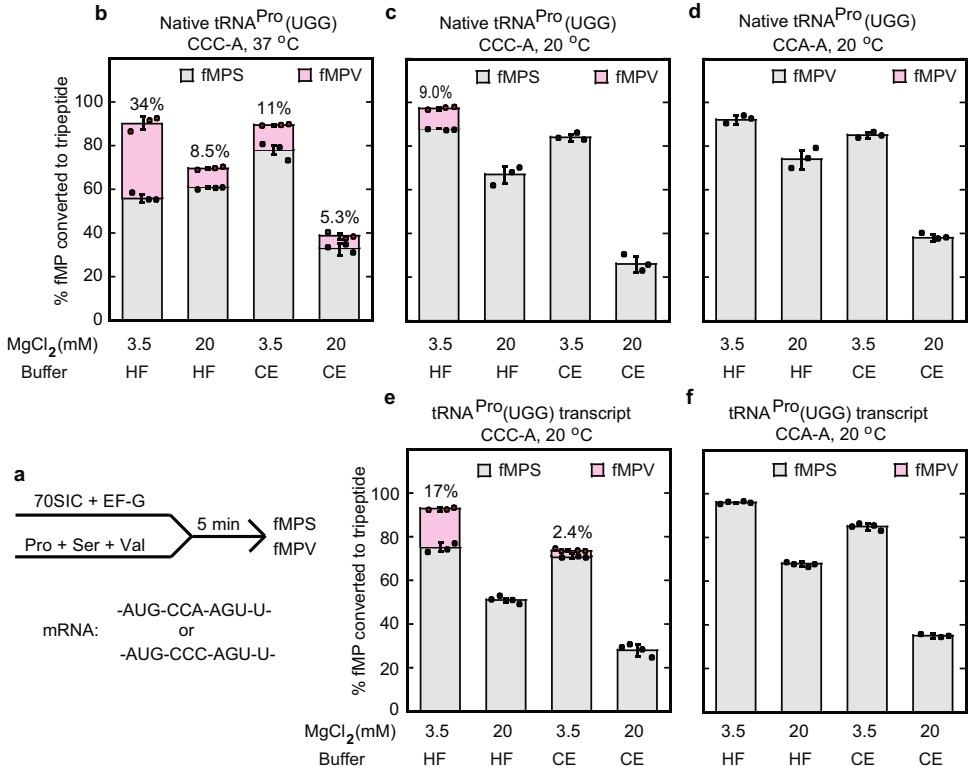

**Fig. 2 +1FS of *E. coli* tRNA^Pro(UGG) in a biochemical functional assay. a** The reaction scheme of the functional assay to measure the yield of +1FS as the fractional conversion of fMP to fMPV. An *E. coli* 70SIC programmed with the non-slippery CCA-A reporter or the slippery CCC-A reporter was mixed with a TC containing EF-Tu-GTP with a native or transcript of tRNA^Pro(UGG), Val-tRNA (*UAC, *U = cmo^5U anticodon), and Ser-tRNA (GCU anticodon) in the HF or CE buffer containing the indicated $MgCl_2$ concentration. Each reaction was quenched after 5 min with 0.5 M KOH. Peptides were resolved by electrophoretic TLC and quantified by phosphor-imaging. **b–f** The fractional conversion of fMP to fMPV (pink) was reported for tRNA^Pro(UGG) in the native-state at the CCC-A codon motif at 37 °C (**b**); in the native-state at the CCC-A codon motif at 20 °C (**c**); in the native-state at the CCA-A codon motif at 20 °C (**d**); in the transcript-state at the CCC-A codon motif at 20 °C (**e**); and in the transcript-state at the CCA-A codon motif at 20 °C (**f**). All data are presented as mean ± SD. The bars in graphs **b–f** are SDs of three independent (*n* = 3) experiments; for datasets CCC-A transcript/HF(20),CE(3.5),CE (20); CCA-A transcript HF(3.5), HF(20),CE(3.5), *n* = 4.

triad is disrupted and provides weaker support for the codon–anticodon helix than in the non-frameshifting structure (Fig. 3c). Structure I-FS therefore reveals that although the codon–anticodon helix is in the normal 0-frame (Fig. 3e and Supplementary Fig. 7a), the cmo^5U34-C3 wobble pairing shifts the 30S dynamics equilibrium toward the open 30S conformation.

Upon peptidyl transfer, the 30S subunit spontaneously rotates as the tRNAs form the hybrid states by shifting their acceptor arms on the large subunit[1,2,42]. Peptidyl-tRNA adopts the A/P or the elbow-shifted A/P* conformations, while deacyl-tRNA forms the P/E state[41]. cryo-EM analyses of the frameshifting complex formed without EF-G (Supplementary Fig. 3) yielded Structure I^rot-FS, in which the 30S subunit body is rotated by ~11° and the 30S head is modestly swiveled by ~4°. Dipeptidyl-tRNA adopts the elbow-shifted A/P* conformation and deacyl-tRNA adopts the P/E conformation (Supplementary Figs. 7b and 8), overall similar to those in cognate 70S complexes formed with different tRNAs[41]. Structure I^rot-FS reveals that the codon–anticodon helix is in the 0-frame and the 30S subunit adopts the open conformation (Supplementary Fig. 8b), similar to that in the non-rotated Structure I-FS (Supplementary Fig. 8c). Together, Structures I-FS and I^rot-FS indicate that no frameshifting occurs upon decoding and peptidyl transfer on the slippery sequence in the A site.

**mRNA frame is shifted in the EF-G-bound structures II-FS and III-FS.** EF-G•GTP binds to the rotated conformation of pre-

translocation ribosomes[2,3,43,44]. Spontaneous reverse rotation of the 30S subunit in the presence of EF-G causes synchronous translocation of tRNA ASLs and mRNA codons within the 30S subunit, resulting in P/P and E/E states upon completion of the rotation[45]. Previous structures of 70S•2tRNA•EF-G complexes captured 30S in rotated states that ranged from ~10 degrees to 0 degrees[43,46–48], revealing early (rotated) and late (non-rotated) stages of translocation. They show that domain IV of EF-G binds next to the translocating peptidyl-tRNA and sterically hinders its return to the A site on the 30S subunit upon reverse subunit rotation[2,49,50].

Our cryo-EM structures reveal two predominant translocation states with EF-G•GDPCP: the partially rotated state (~5°) and the nearly non-rotated state (~1°; relative to the non-rotated pre-translocation structure I) (Figs. 4 and 5). The non-frameshifting structures II and III closely resemble previously described mid-translocated[47,48] (Fig. 4a–c) and post-translocated[46] structures (Fig. 5a, b) formed with antibiotics. In the partially rotated state, the head of the 30S subunit is swiveled by ~16°, so the 30S beak is closer to the 50S subunit (Fig. 4a). The head swivel is coupled with tRNA ASL and mRNA translocation on the small subunit, allowing gradual translocation first relative to the 30S body and then the 30S head[51]. In the head-swiveled Structure II, dipeptidyl fMP-tRNA^Pro is between the A and P sites of the 30S subunit (Fig. 4b). Here, the anticodon is only ~4 Å away from the P site of the body domain (measured at cmo^5U34), but it remains near the A site of the head domain due to the movement of the head in the

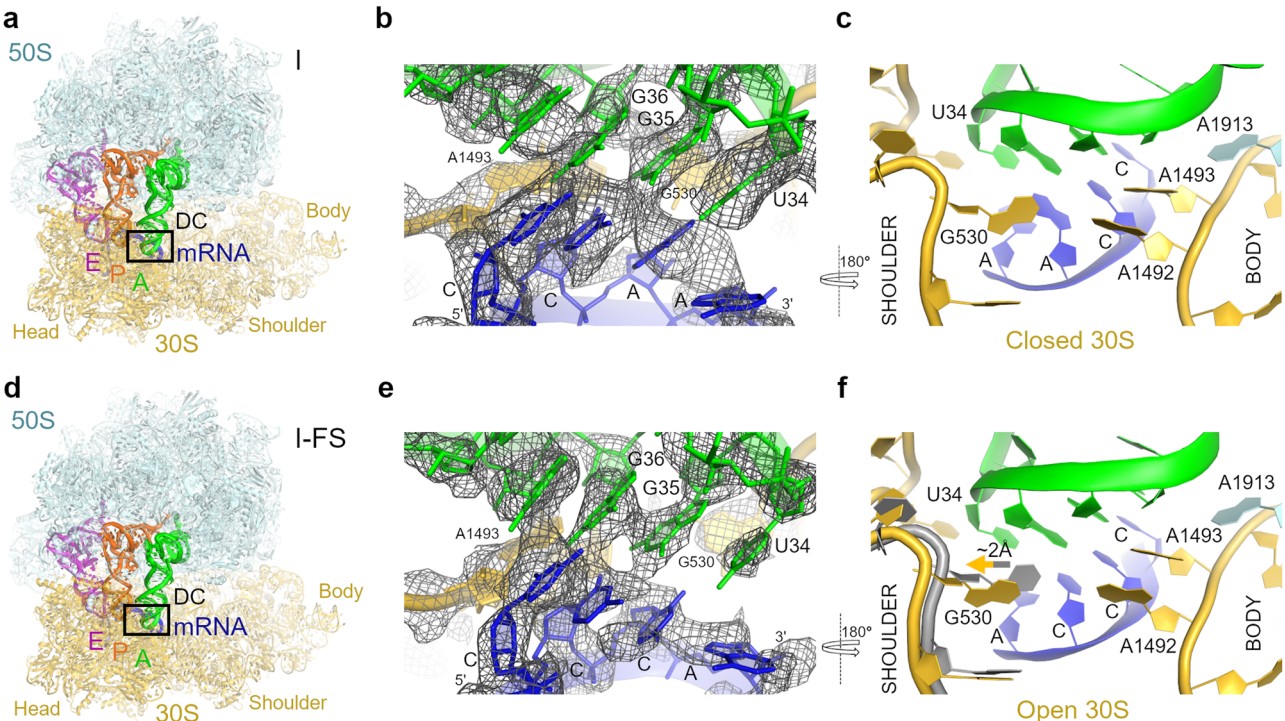

**Fig. 3 Cryo-EM structures of pre-translocation 70S formed with fMet-tRNA^fMet (P site) and Pro-tRNA^Pro (A site). a** Overall view of the 70S structure with non-frameshifting mRNA (CCA-A; Structure I). Weaker density in the E-site than in the A and P sites suggests partial occupancy of E-tRNA ("Methods"). **b** Cryo-EM density (gray mesh) for codon–anticodon interaction between non-frameshifting mRNA and tRNA^Pro in the A site of Structure I. The view approximately corresponds to the boxed decoding center region (DC) in (**a**). The map was sharpened with a B-factor of -80 Å² and is shown at 2.5 σ. **c** Decoding center nucleotides G530 (in the shoulder region) and A1492-A1493 (in the body region) stabilize the codon–anticodon helix in Structure I. **d** Overall view of the 70S structure with the slippery mRNA (CCC-A; Structure I-FS). Weaker density in the E-site than in the A and P sites suggests partial occupancy of E-tRNA (see "Methods"). **e** Cryo-EM density (gray mesh) for codon–anticodon interaction between the slippery mRNA codon and tRNA^Pro in Structure I-FS. The map was sharpened with a B-factor of -80 Å² and is shown at 2.5 σ. **f** Partially open conformation of the 30S subunit due to the shifted G530 (in the shoulder region) in Structure I-FS relative to that in Structure I (16S shown in gray). Structural alignment was obtained by superposition of 16S ribosomal RNAs (rRNAs). In all panels, the 50S subunit is shown in cyan, 30S subunit in yellow, mRNA in blue, tRNA^Pro in green, tRNA^fMet in orange and sub-stoichiometric E-site tRNA in magenta.

direction of translocation. The acceptor arm is in the P site of the 50S subunit. Thus, the tRNA conformation is similar to the previously described chimeric ap/P conformation[47] (denoting the anticodon at the A site of the 30S head and near the P site of the 30S body (ap), and the acceptor arm in the P site of the 50S subunit (P)).

The nearly non-rotated Structure III features a small head swivel (~1°) and dipeptidyl-tRNA in the P site (Fig. 5a, b), resembling the non-rotated post-translocation ribosome[46]. Both the dipeptidyl-tRNA^Pro and the deacylated tRNA^fMet are base paired with their respective mRNA codons in the P and E sites, respectively. In both structures II and III, domain IV of EF-G interacts with the ASL of the dipeptidyl-tRNA and the cognate CCA codon (Figs. 4b, c and 5b), consistent with the role of EF-G in stabilizing the codon–anticodon helix during translocation[47] and after arrival of the codon–anticodon helix at the P site[46]. As in previous EF-G-bound structures with a catalytically inactive EF-G or with GTP mimics[48,52–54], the switch loops in EF-G domain I are well resolved in Structures II and III, consistent with stabilization of the GTPase by GDPCP (Supplementary Fig. 9).

By contrast, EF-G•GDPCP mediates frameshifting on the frameshift-prone CCC-A mRNA motif. In the mid-translocated Structure II-FS, the dipeptidyl-tRNA^Pro (Supplementary Fig. 7c) pairs with the mRNA in the +1-frame (C²CA⁴) between the A and P sites of the 30S subunit (Fig. 4d–f). Here, clearly resolved density demonstrates base-pairing of cmo⁵U34 of tRNA^Pro with A4 of the mRNA (Fig. 4e), although the cmo⁵ moiety of U34 is

poorly resolved in this and other structures likely due to its conformational dynamics. The neighboring deacylated tRNA^fMet is bound to the AUG codon near the E-site. Thus, +1FS results in a bulged mRNA nucleotide C1 between the E and P sites (Fig. 4e, g, i). C1 is sandwiched between the guanosine of the AUG codon and G926 of 16S rRNA. This stabilization allows mRNA compaction and accommodation of four mRNA nucleotides in the E-site, which normally accommodates three nucleotides. Due to frameshifting, tRNA^fMet and tRNA^Pro are shifted away from each other; they are moved by 4 Å and 3 Å from their positions in the non-frameshifting Structure II, respectively (Fig. 4g). The shift of tRNA^Pro is compensated by the shift of EF-G loop II (His584-Asp587), critical for fast translocation[55,56], whereas the rest of EF-G domain IV including loop I (Ser509-Gly511) is placed similarly to that in the non-frameshifting complex (Fig. 4h).

Previous crystallographic work suggested that the 16S rRNA nucleotides C1397 and A1503, which flank the A and E sites, respectively, prevent mRNA slippage by interacting with the bases of translocating mRNA[48,57]. These two nucleotides are part of the central region of the 30S head that is stabilized by numerous interactions, including the conserved 1399-1504 Watson–Crick base pair formed by nucleotides neighboring the "stoppers" C1397 and A1503. Our structures indicate that the positions and conformations of this head region, including C1397 and A1503, are nearly identical between the non-frameshifting Structure II and the frameshifted Structure II-FS (Supplementary Fig. 9). Thus, the compact and frameshifted mRNA can be

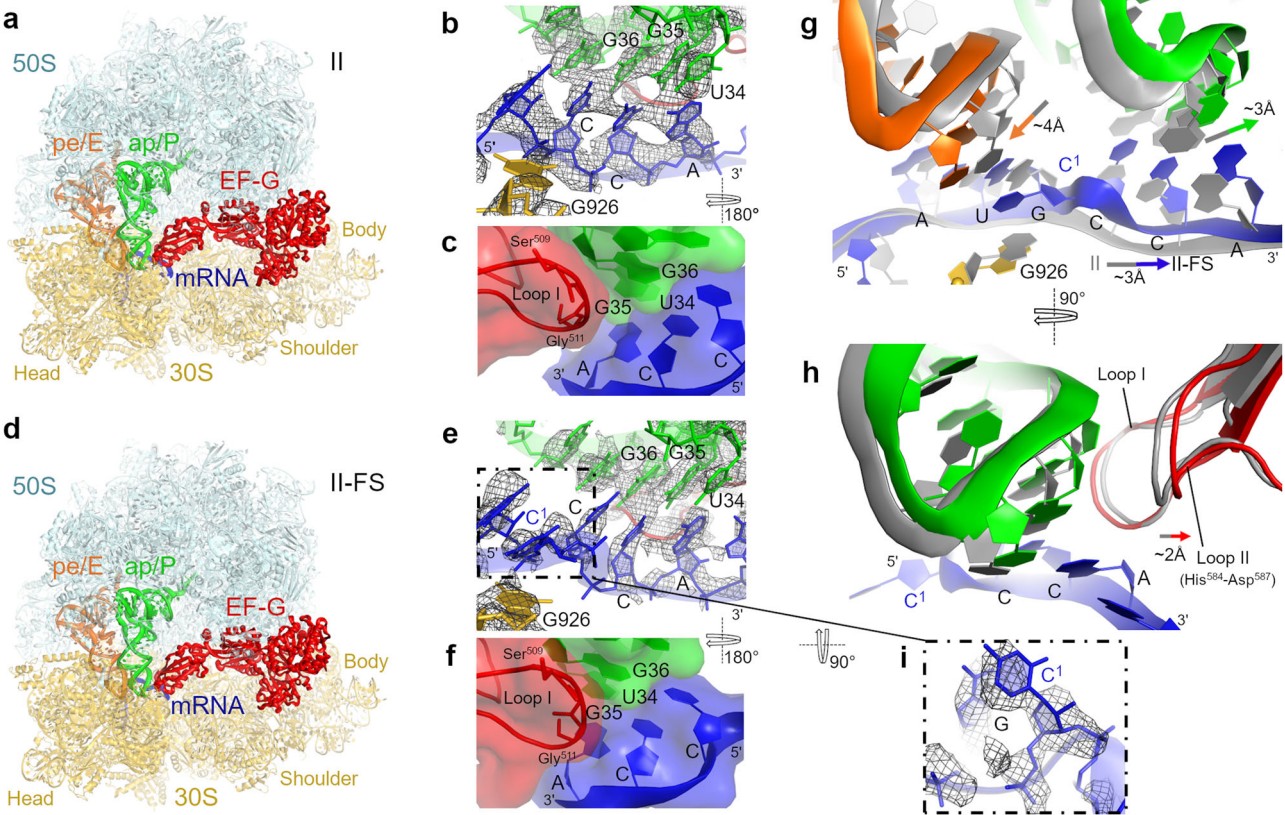

**Fig. 4 Cryo-EM structures of mid-translocation states formed with EF-G•GDPCP. a** Overall view of mid-translocation Structure II with the non-frameshifting mRNA. **b** Cryo-EM density (gray mesh) of the non-frameshifting tRNA$^{Pro}$ and mRNA codon near the P site. The map is sharpened by applying the B-factor of -80 Å$^2$ and is shown with 2.5 σ. **c** Interaction of the EF-G loop I (Ser509-Gly511, red) with the codon–anticodon helix (space-filling surface and cartoon representation). **d** Overall view of mid-translocation Structure II-FS with the frameshifting mRNA. **e** Cryo-EM density (gray mesh) of the frameshifting tRNA$^{Pro}$ and mRNA codon near the P site. The map is sharpened by applying the B-factor of -80 Å$^2$ and is shown with 2.5 σ. Note the unpaired and bulged C$^1$ nucleotide in the mRNA, also shown in (**i**). **f** Interaction of the EF-G domain IV loop I (Ser509-Gly511) with the codon–anticodon helix of the frameshifting mRNA (compare to **c**). **g** Differences in positions of tRNA$^{Pro}$ (green) and tRNA$^{fMet}$ (orange) in the frameshifting structure II-FS relative to those in the non-frameshifted structure II (gray). **h** Adjustment of loop II of domain IV of EF-G (red) to accommodate the shifted position of tRNA$^{Pro}$ (green) in the frameshifting structure II-FS relative to those in structure II (gray). Structural alignments were performed by superposition of 16S rRNAs. **i** Close-up view of cryo-EM density for bulged C1 in Structure II-FS (also shown in **e**). The ribosomal subunits, tRNAs, and mRNA are colored as in Fig. 3, EF-G is shown in red.

accommodated in the ribosomal mRNA tunnel during transloca-tion without perturbing the conformations of the head nucleotides.

In the nearly translocated non-rotated Structure III-FS (Fig. 5c), the +1-frame CCA codon and dipeptidyl-tRNA$^{Pro}$ are in the P site, while C1 and the AUG codon with the deacylated tRNA$^{fMet}$ are in the E-site (Fig. 5d). To accommodate C1 in the E-site, the E-site AUG codon and tRNA$^{fMet}$ are shifted by up to 3 Å (Fig. 5e). While most of tRNA$^{fMet}$ is well resolved, poor density for the ASL indicates destabilization of E-site codon-anticodon interactions. Weak C1 density suggests that C1 is detached from G926, which instead hydrogen-bonds with the phosphate group of the first nucleotide of the P-site codon (Fig. 5d, e). The P-site codon and tRNA$^{Pro}$ are positioned nearly identically to those in the non-frameshifting Structure III (Fig. 5f). Thus, the frame-shifted mRNA and peptidyl-tRNA are placed at the canonical P-site position at the end of the translocation trajectory, preparing the ribosome for the next elongation cycle on the new +1-frame of the mRNA.

**Structural mechanism of +1 frameshifting.** cryo-EM structures in this work provide the long-sought snapshots of +1 FS (Fig. 6), which are consistent with the recent biophysical work[18] and other

studies suggesting that frameshifting occurs during EF-G-catalyzed translocation. The use of the native *E. coli* tRNA$^{Pro}$ (UGG) and visualization of EF-G-bound structures distinguishes this work from previous structural studies that were based on +1FS suppressor tRNAs with an expanded anticodon loop[26,31–33] or on frameshifting-like complexes with a single tRNA[58,59]. To obtain a complete +1-FS-prone elongation complex with two tRNAs required for translocation, we placed a frameshifting mRNA sequence C$^1$CC-A$^4$ and tRNA$^{Pro}$(UGG)[17] in the A site. The frameshifting ribosome complex therefore contains a wobble cmo$^5$U34-C3 pair upon binding of tRNA$^{Pro}$ to the C$^1$CC-A$^4$ sequence (Structure I-FS). Although the downstream A4 would have been a more favorable base-pairing partner for cmo$^5$U34 of tRNA$^{Pro}$(UGG), there is no frameshifting upon decoding and peptidyl transfer (Structures I-FS and I$^{rot}$-FS). Thus, the +1FS-prone pre-translocation complex maintains the 0-frame anticodon–codon pairing resembling that in canonical elonga-tion complexes[40] and crystal structures with +1FS suppressor tRNAs[31–33]. However, unlike the 0-frame complexes containing the cmo$^5$U34-A3 base pair (Structure I, Fig. 6a) or previous complexes with suppressor tRNAs[31–33], structure I-FS features an open 30S subunit, resembling transient decoding intermediates[38]. Here, G530 of 16S rRNA is shifted from its canonical position near the second base pair of the codon–anticodon helix[37], thus

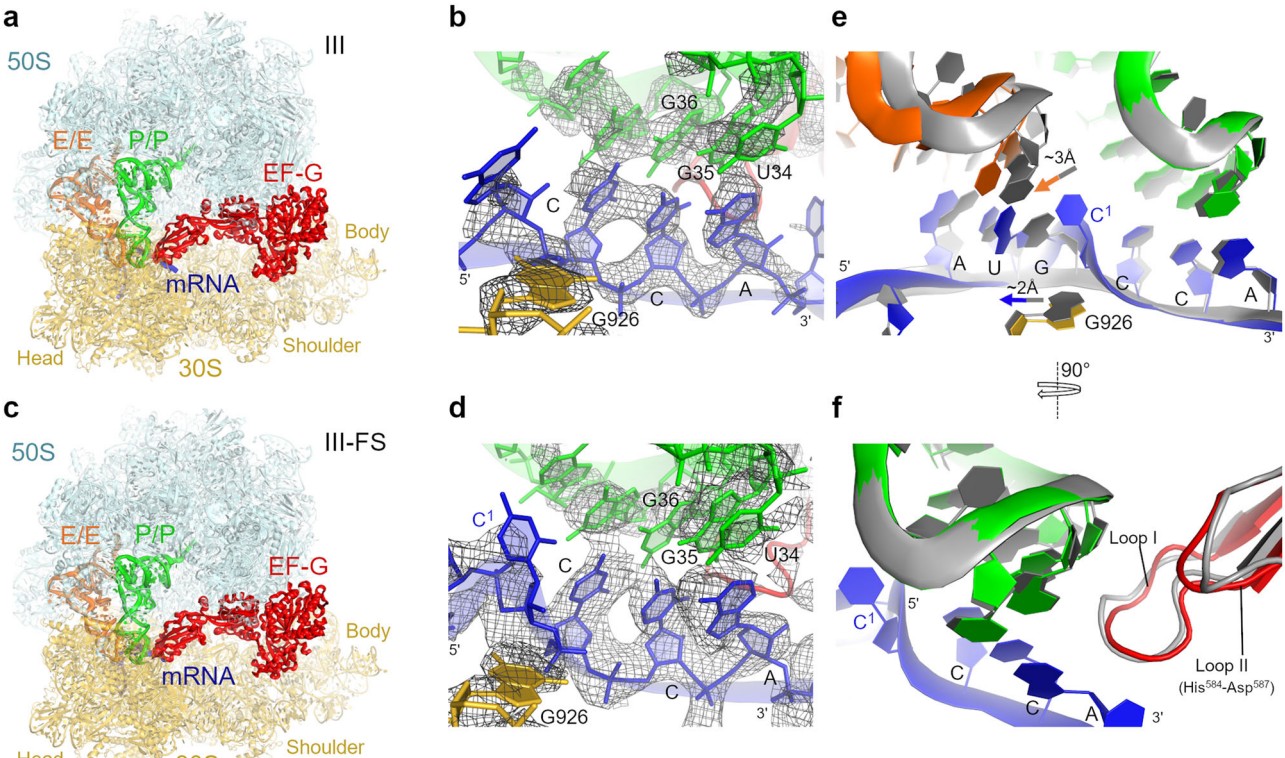

**Fig. 5 Cryo-EM structures of near post-translocation states formed with EF-G•GDPCP. a** Overall view of the near-post-translocation Structure III with the non-frameshifting mRNA. **b** Cryo-EM density (gray mesh) of the non-frameshifted tRNA$^{Pro}$ and mRNA codon at the P site. The map was sharpened by applying the B-factor of -80 Å$^2$ and is shown at 2.5 σ. **c** Overall view of the near post-translocation Structure III-FS with the frameshifting mRNA. **d** Cryo-EM density (gray mesh) of the frameshifted tRNA$^{Pro}$ and mRNA codon at the P site. The map was sharpened by applying the B-factor of -80 Å$^2$ and is shown at 2.5 σ. **e** Comparison of mRNA and tRNA positions in the nearly translocated frameshifted (colored, III-FS) and non-frameshifted (gray, III) complexes. **f** Positions of loop II of EF-G (red) and tRNA$^{Pro}$ (green) in the Structures III-FS and III (gray). Structural alignments were performed by superposition of 16S rRNAs. The color scheme is as in Fig. 4.

possibly destabilizing the labile three-base-pair codon–anticodon helix containing the cmo$^5$U34-C3 pair upon efficient accommodation and peptidyl transfer[18]. The pre-translocation ribosome therefore appears to pre-dispose tRNA$^{Pro}$ for sliding from its near-cognate codon CCC in the 0-frame to the cognate CCA codon in the +1-frame (Fig. 6b). Limited space in the A site, however, restricts the codon–anticodon dynamics and prevents slippage in this pre-translocation state.

In contrast to pre-translocation complexes, the mid-translocation complex with EF-G, and the highly swiveled 30S head features tRNA$^{Pro}$ base paired with the +1-frame C$^2$CA$^4$ codon near the P site of the body and the A site of the head (Structure II-FS). This suggests that the ribosome switches to the +1-frame when tRNA$^{Pro}$ and mRNA move from the decoding center, and that frameshifting is accomplished by the intermediate of EF-G-catalyzed translocation, at which the tRNA is nearly translocated along the 30S body. The complex remains frameshifted till the completion of translocation when tRNA$^{Pro}$ is in the P site relative to both the body and head due to the reverse head swivel (Structure III-FS). Our work therefore suggests a structural mechanism (Fig. 6b), in which non-canonical pairing in the pre-translocation complex sets the stage for frameshifting by opening the 30S subunit and promoting frameshifting during EF-G-catalyzed translocation.

Our observation of destabilization of the pre-translocation complex and of EF-G-bound frameshifting structures is consistent with the high efficiency of +1FS on the mismatched CCC-A frameshifting codon motif shown in vitro[17] and in cells (Fig. 1c). Other frameshifting sequences exist, however, which contain fully

complementary codon–anticodon interactions in the 0- and +1-frames, including the CCC-C sequence decoded by tRNA$^{Pro}$ (GGG)[22], as demonstrated in Fig. 1d. In these cases, the pre-translocation complex most likely samples the canonical closed 30S conformation, in which the codon–anticodon helix is stabilized by the decoding center (as in Structure I). This frame stabilization must at least in part account for the lower efficiency of frameshifting on such sequences[17] (Fig. 1d). Nevertheless, the low frequency with which +1FS occurs with such sequences indicates that the tRNA-mRNA interactions can be stochastically destabilized during translocation, when the 30S subunit, tRNAs, and mRNA rearrange. Indeed, recent 70S structures obtained without EF-G demonstrate mRNA frame destabilization upon 30S head swiveling. In a frameshift-like complex featuring a single tRNA and a swiveled 30S head, the bulged nucleotide between the E and P-site codons is stabilized by G926[59], similarly to that in Structure II-FS. Furthermore, a recent crystal structure of a non-frameshifting complex with two tRNAs and swiveled 30S head revealed perturbation of the codon–anticodon interactions in the P site, despite full complementarity of the P-site tRNA with the 0-frame codon[57]. While tRNA-mRNA pairing is unstable during head swiveling, EF-G maintains the reading frame in non-frameshifting complexes by interacting with both the tRNA anticodon and mRNA codon along the translocation trajectory (Structures II and III). By contrast, in the frameshifting-prone complexes, EF-G fails to support the codon–anticodon interactions that are transiently destabilized during translocation along with the 30S subunit (such as the CCC-A motif in this study) and allows slippage into the +1 frame that is fully complementary to

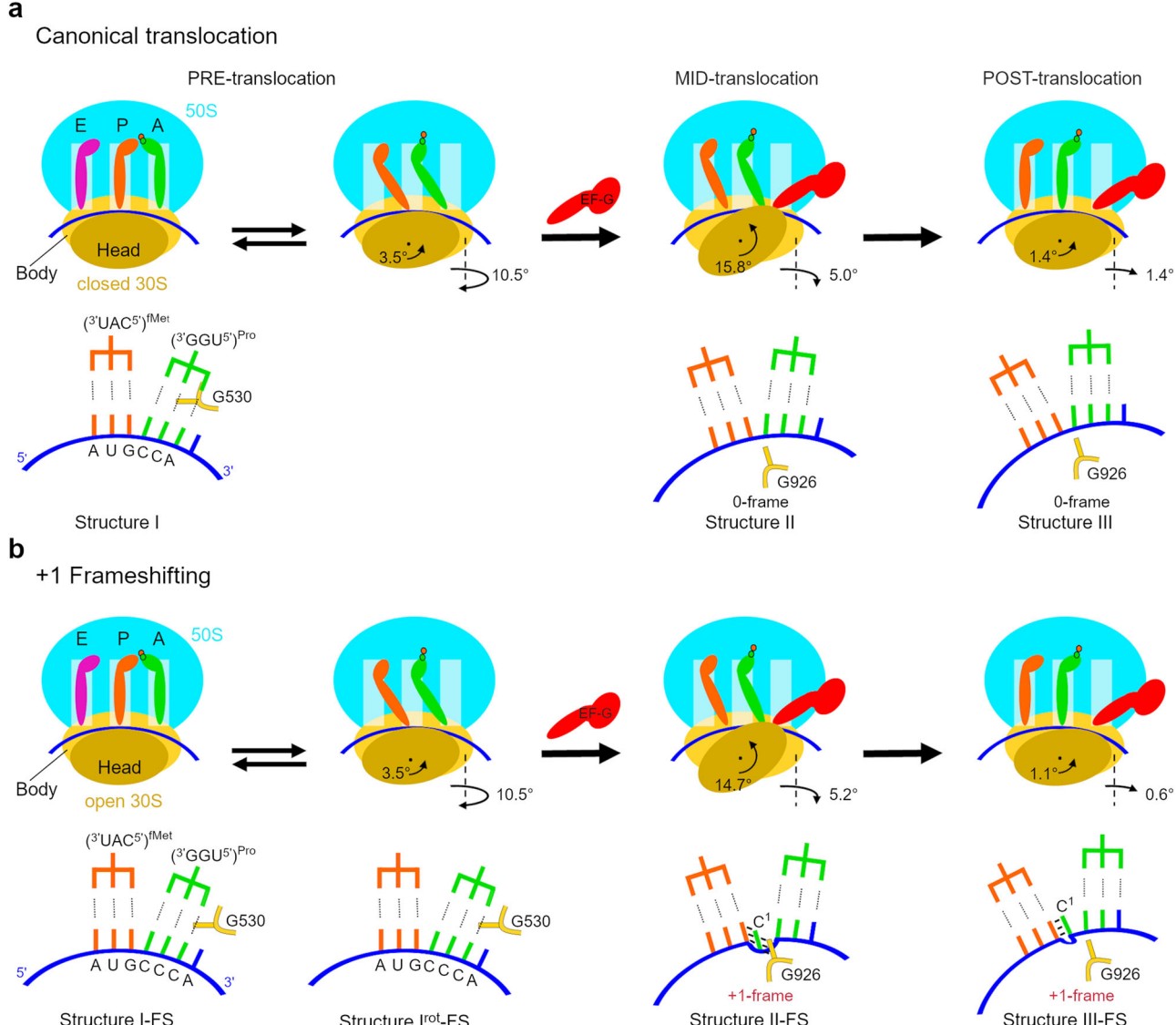

**Fig. 6 The mechanism of + 1 frameshifting. a** Schematic of canonical ribosomal translocation by EF-G and ribosome rearrangements. **b** Schematic of ribosomal translocation by EF-G resulting in +1 frameshifting. The second rows in **a**, **b** show local rearrangements of mRNA-tRNA and positions of the decoding-center nucleotide G530 and P-site nucleotide G926 of the 30S subunit. The color scheme is as in Fig. 4.

tRNA. Upon slippage, EF-G and 30S residues can stabilize the new frame at the final stages of translocation (Structures II-FS and III-FS; Fig. 6b). We cannot exclude an alternative scenario, in which cellular frameshifting occurs during or after EF-G•GDP dissociation from the ribosome in a head-swiveled conformation[60]. In the cryo-EM sample equilibrated with EF-G•GDPCP, frameshifting could have initiated on fully translocated ribosomes (as in III-FS), which spontaneously reverted to the frameshifted mid-translocation states with stalled EF-G (II-FS). Either scenario is consistent with the prevalence of frameshifting in the stalled cryo-EM structures, which contrasts the small fraction of frameshifting during dynamic translocation with EF-G•GTP (Fig. 2), emphasizing the key role of timely association and dissociation of EF-G in mRNA frame maintenance.

## Methods

**Cell-based assays for +1FS**. The *E. coli lacZ* plasmid in pKK223-3, which we developed previously[22], was modified by QuikChange mutagenesis to contain a CCX-X motif following the start codon AUG of the IPTG-inducible reporter gene. Control plasmids containing the in-frame insertion of CCA or CCC were made in parallel. Primers used for mutagenesis to insert a slippery or non-slippery motif in the lacZ plasmid are listed in Supplementary Table 1. The measured β-gal activity from a reporter was normalized by the activity of the corresponding CCA or CCC control plasmid to calculate the +1FS frequency of the reporter. To determine the effect of m[1]G37 on the frequency of +1FS, each reporter and the corresponding control plasmid was expressed in an *E. coli trmD-KO*/JM109 strain, where *trmD* was eliminated from the chromosome and cell survival was maintained by arabinose (Ara)-controlled expression of the human counterpart *trm5* from the pACYC plasmid. This *E. coli trmD-KO*/JM109 strain was made by transducing the P1 lysate of a previously described *E. coli trmD-KO*/MG1655[22] into JM109. To determine the contribution of the tRNA[Pro](GGG) isoacceptor to +1FS, a *proL*-deletion mutant of *E. coli trmD-KO*/JM109 was made so that only the *proM* tRNA[Pro](UGG) isoacceptor was active for decoding the CCA-A or CCC-A codon motif in a reporter. This *proL*-deletion mutant of *E. coli trmD-KO*/MG1655 was made by replacing *proL* on the chromosome with a Kan marker using the P1 lysate of a previously described *E. coli proL-KO* strain[18], followed by removing the Kan marker with pCP20.

To measure the frequency of +1FS, the *E. coli trmD-KO*/JM109 strain maintained by *trm5* was transformed with a *lacZ* reporter plasmid. A single colony of each strain was grown overnight in Luria-Bertani (LB) medium in the presence of 0.2% Ara at 37 °C, then inoculated 1:100 to fresh LB with or without 0.2% Ara to generate an m[1]G37+ or m[1]G37− condition, respectively. After 1-h growth at 37 °C, during which Trm5 was depleted and m[1]G37 level reduced in the m[1]G37− condition, 0.4 mM IPTG was added to turn on the *lacZ* gene, and cells were grown for additional 4 h at 37 °C. Cells were harvested and the β-gal activity was measured

as a Miller Unit[22]. The +1FS efficiency (%) was calculated by determining the β-gal activity of a reporter construct relative to that of the corresponding in-frame reference construct. We used CCA for CCA-A and CCC for CCC-A and CCC-C for reference.

**Biochemical assays for +1FS.** Two mRNAs used in the biochemical study were prepared by in vitro transcription with T7 RNA polymerase to place the test sequence CCX-X after the start codon. The non-slippery mRNA [5′-GGG AAG GAG GUA AAA AUG CCA AGU UAU AAG CAC CAC CAC CAC CAC CAC CAC] contained the non-slippery CCA-A codon motif after the AUG start codon, whereas the slippery mRNA contained the slippery CCC-A motif in an otherwise identical sequence context. Native E. coli tRNA[Pro](UGG) was isolated from cells over-expressing proM by affinity purification[18]. The transcript of tRNA[Pro](UGG) was generated by T7 transcription, first as a precursor with a self-cleaving 5′-ribozyme, which was processed to release the mature form and gel purified. For other tRNAs, native tRNA[Val](*UAC, where *U = cmo5U) and tRNA[fMet](CAU) were overexpressed in E. coli and isolated from a total tRNA pool, whereas tRNA[Ser] (GCU) was made by in vitro transcription. Each elongator tRNA was charged by its cognate aminoacyl-tRNA synthetase, followed by phenol extraction, ethanol precipitation, and storage at −70 °C in 25 mM NaOAc, pH 5.0. The extent of charging was monitored by doping the reactions with tritiated amino acid and by measuring for incorporation of the tritiated amino acid into tRNA after centrifugation through a gel filtration spin column. Charging of initiator tRNA[fMet] was done in the presence of 35S-methionine. Formylation was carried out simultaneously with charging by including formyl transferase and the formyl donor 10-formyltetrahydrofolate[61]. Tight-coupled 70S ribosomes and His-tagged translation factors were isolated from E. coli MRE600 cells and stored at −70 °C.

E. coli 70SIC was formed by incubating a 70S ribosome for 25 min at 37 °C with an mRNA containing CCC-A or CCA-A next to the AUG start codon, 35S-fMet-tRNA[fMet], initiation factors IF1, IF2 and IF3, and EF-G in the HF[3.5] buffer (3.5 mM MgCl₂, 0.5 mM spermidine, 1 mM dithiothreitol, 30 mM KCl, 70 mM NH₄Cl, 50 mM Tris-HCl, pH 7.5) or the cryo-EM[3.5] buffer (3.5 mM MgCl₂, 0.05 mM spermine, 2 mM spermidine, 6 mM 2-mercaptoethanol, 120 mM NH₄Cl, 20 mM HEPES-KOH, pH 7.5) supplemented with 0.5 mM GTP. TCs were formed by first incubating EF-Tu at 37 °C for 15 min in the HF[3.5] or cryo-EM[3.5] buffer with 0.5 mM GTP, followed by the addition of native or transcript Pro-tRNA[Pro](UGG), transcript Ser-tRNA[Ser](GCU), and native Val-tRNA[Val](*UAC) for 15 min in an ice bath. Prior to mixing equal volumes of the TC and 70SIC, the MgCl₂ concentration of each was kept at 3.5 mM or adjusted upward to 7–20 mM as indicated. Reactions were carried out at 20 or 37 °C and contained final concentrations of 0.4 μM ribosome, 0.25 μM 35S-fMet-tRNA[iMet], 0.5 μM mRNA (CCC-A or CCA-A), 0.5 μM each IFs I, II, and III, 3 μM EF-Tu, 0.5 μM tRNA[Pro](UGG), 0.75 μM each Ser- and Val-tRNA, 2 μM EF-G, and 0.5 mM GTP in HF or cryo-EM buffer with 3.5, 7.0, 10, 13.5, 17, or 20 mM MgCl₂ as indicated. Reaction aliquots were quenched in 0.5 M KOH and were kept at 37 °C for 30 min before loading 0.8 μL of each onto a 20 cm cellulose thin-layer-chromatography (TLC) sheet. Electrophoresis at 800 V for 2¼ h in PYRAC buffer resolved fMP, fMPV, and fMPS peptides when samples were loaded onto an origin 15 cm from the anode end of the sheet. The dried TLC sheet was visualized by phosphor-imaging and spots were quantified using ImageJ[62].

**Preparation of EF-G and ribosomal subunits for cryo-EM.** The gene encoding full-length E. coli EF-G (704 aa, C-terminally His₆-tagged) in pET24a+ plasmid (Novagen, kanamycin resistance vector) was transformed into an E. coli BLR/DE3 strain. Cells with the plasmid were cultured in LB medium with 50 μg mL⁻¹ kanamycin at 37 °C until the OD₆₀₀ reached 0.7–0.8. Expression of EF-G was induced by 1 mM IPTG (Gold Biotechnology Inc., USA), followed by cell growth for 9 h at 16 °C. The cells were harvested, washed, and resuspended in buffer A (50 mM Tris-HCl pH 7.5, 50 mM NH₄Cl, 10 mM MgCl₂, 5% glycerol, 10 mM imidazole, 6 mM β-mercaptoethanol (βME), and a cocktail of protease inhibitors (complete Mini, EDTA-free protease inhibitor tablets, Sigma-Aldrich, USA). The cells were disrupted with a microfluidizer (Microfluidics, USA), and the soluble fraction was collected by centrifugation at 36,000 × g (JA-20 rotor; 18,000 rpm) for 50 min and filtered through a 0.22 μm pore size sterile filter (CELLTREAT Scientific Products, USA).

EF-G was purified in three steps. The purity of the protein after each step was verified by 12% SDS-PAGE stained with Coomassie Brilliant Blue R 250 (Sigma-Aldrich). First, affinity chromatography with Ni-NTA column (Nickel-nitrilotriacetic acid, 5 ml HisTrap, GE Healthcare) was performed using FPLC (Äkta explorer, GE Healthcare). The soluble fraction of cell lysates was loaded onto the column equilibrated with buffer A and washed with the same buffer. EF-G was eluted with a linear gradient of buffer B (buffer A with 0.25 M imidazole). Fractions containing EF-G were pooled and dialyzed against buffer C (50 mM Tris-HCl pH 7.5, 100 mM KCl, 10 mM MgCl₂, 0.5 mM EDTA, 6 mM βME, and the cocktail of protease inhibitors). The protein then was purified by ion-exchange chromatography through a HiPrep FF Q-column (20 mL, GE Healthcare; FPLC). After the column was equilibrated and washed with Buffer C, the protein was loaded in Buffer C and eluted with a linear gradient of Buffer D (Buffer C with 0.7 M KCl). Finally, the protein was dialyzed against 50 mM Tris-HCl pH 7.5, 100 mM KCl, 10 mM MgCl₂, 0.5 mM EDTA, 6 mM βME, and purified using size-exclusion

chromatography (Hiload 16/600 Superdex 200 pg column, GE Healthcare). The fractions of the protein were pooled, buffer exchanged (25 mM Tris-HCl pH 7.5, 100 mM NH₄Cl, 10 mM MgCl₂, 0.5 mM EDTA, 6 mM βME, and 5% glycerol) and concentrated with an ultrafiltration unit using a 10-kDa cutoff membrane (Millipore). The concentrated protein was flash-frozen in liquid nitrogen and stored at −80 °C.

70S ribosomes were prepared from E. coli (MRE600)[63] and stored in the ribosome-storage buffer (20 mM Tris-HCl (pH 7.0), 100 mM NH₄Cl, 12.5 mM MgCl₂, 0.5 mM EDTA, 6 mM βME) at −80 °C. Ribosomal 30S and 50S subunits were purified using a sucrose gradient (10–35%) in a ribosome-dissociation buffer (20 mM Tris-HCl (pH 7.0), 500 mM NH₄Cl, 1.5 mM MgCl₂, 0.5 mM EDTA, and 6 mM βME). The fractions containing 30S and 50S subunits were collected separately, concentrated, and stored in the ribosome-storage buffer at −80 °C.

**Preparation of charged tRNAs, and mRNA sequences for cryo-EM.** E. coli tRNA[fMet] was purchased from Chemical Block. Native E. coli tRNA[Pro](UGG) (proM tRNA) was overexpressed in E. coli from an IPTG-inducible proM gene carried by pKK223-3. Total tRNA was isolated using differential centrifugation and proM tRNA was isolated using a complementary biotinylated oligonucleotide attached to streptavidin-sepharose yielding approximately 40 nmoles proM tRNA from 1 liter of culture. E. coli tRNA[Pro] (UGG) (10 μM) was aminoacylated in the charging buffer (50 mM HEPES-KOH pH 7.5, 50 mM KCl, 10 mM MgCl₂, 10 mM DTT) in the presence of 40 μM L-proline, 2 μM prolyl-tRNA synthetase (ProRS), 0.625 mM ATP and 15 μM elongation factor EF-Tu (purified as in our recent work[38]. The mixture was incubated for 10 min at 37 °C. To stabilize the charged Pro-tRNA[Pro] and form the ternary complex for the elongation reaction 0.25 mM GTP was added to the mixture. The mixture was incubated for 3 min at 37 °C.

mRNAs containing the Shine–Dalgarno sequence and a linker to place the AUG codon in the P site were synthesized by IDT. The frameshifting mRNA contains the sequence 5′-GGC AAG GAG GUA AAA AUG CCC AGU UCU AAA AAA AAA, and the non-frameshifting mRNA contains the sequence 5′-GGC AAG GAG GUA AAA AUG CCA AGU UCU AAA AAA AAA AAA.

**Preparation of 70S translocation complexes with or without EF-G•GDPCP.** 70S•mRNA•fMet-tRNA[fMet]•Pro-tRNA[Pro](UGG)•EF-G•GDPCP complexes were prepared as follows, separately for the slippery and non-slippery mRNAs. In each, 0.33 μM 30S subunits (all concentrations specified for the final solution) were pre-activated at 42 °C for 5 min in the ribosome-reconstitution buffer (20 mM HEPES-KOH pH 7.5, 120 mM NH₄Cl, 20 mM MgCl₂, 2 mM spermidine, 0.05 mM spermine, 6 mM βME). These activated 30S subunits were added with 0.33 μM 50S subunits with 1.33 μM mRNA and incubated for 10 min at 37 °C. Subsequently, 0.33 μM fMet-tRNA[fMet] was added and the solution was incubated for 3 min at 37 °C, to form the 70S complex with the P-site tRNA.

Pro-tRNA[Pro] (UGG) (0.33 μM), EF-Tu (0.5 μM), and GTP (8.3 μM) were added to the solution and incubated for 10 min at 37 °C to form the A-site bound 70S complex. Next, EF-G (5.3 μM) and GDPCP (0.66 mM) were added and incubated for 5 min at 37 °C, then cooled down to room temperature, resulting in 70S translocation complexes with EF-G•GDPCP.

Pre-translocation 70S•mRNA•fMet-tRNA[fMet]•Pro-tRNA[Pro](UGG) complex that yielded Structure I[rot]-FS was prepared with the slippery mRNA as above excluding the addition of EF-G and GDPCP.

**Cryo-EM and image processing.** QUANTIFOIL R 2/1 grids with the 2-nm carbon layer (Cu 200, Quantifoil Micro Tools) were glow discharged with 25 mA with negative polarity for 60 s (15 mA for the rotated pre-translocation complex without EF-G•GDPCP) in a PELCO easiGlow glow discharge unit. Each complex (2.5 μL) was separately applied to the grids. Grids were blotted at blotting force 9 for 4 s at 5 °C, 95% humidity, and plunged into liquid ethane using a Vitrobot MK4 (FEI). Grids were stored in liquid nitrogen.

cryo-EM data were collected at the cryo-EM Center of the University of Massachusetts Medical School (Worcester, MA, USA). For the non-frameshifting 70S•mRNA(CCA-A)•fMet-tRNA[Pro](UGG)•EF-G•GDPCP translocation complex, a dataset containing 62,716 particles was collected as follows. A total of 1041 movies were collected on a Titan Krios (FEI) microscope (operating at 300 kV) equipped with the K2 Summit camera system (Gatan), with −0.8 to −2.0 μm defocus. Multi-shot data collection was performed by recording four exposures per hole, using SerialEM with a beam-image shift[64]. Each exposure was acquired with continuous data streaming at 36 frames per 7.2 s, yielding a total dose of 47.5 e⁻/Å². The dose rate was 7.39 e⁻/upix/s at the camera. The nominal magnification was 130,000 and the calibrated super-resolution pixel size at the specimen level was 0.525 Å. The movies were motion-corrected and frame averages were calculated using all 36 frames within each movie after multiplying by the corresponding gain reference in IMOD[65]. During motion correction in IMOD the movies were binned to pixel size 1.05 Å (termed unbinned or 1×binned). cisTEM[66] was used to determine defocus values for each resulting frame average and for particle picking. All movies were used for further analysis after inspection of the averages and the power spectra computed by CTFFIND4 within cisTEM. The stack and particle parameter files were assembled in cisTEM with the binnings of 1×, 2× and 4× (box size of 400 for a unbinned stack). Data classification is

summarized in Supplementary Fig. 1. FREALIGNX was used for all steps of particle alignment, refinement, and final reconstruction steps, and FREALIGN v9.11 was used for 3D classification steps[67]. Conversion of parameter file from FREALIGNX to FREALIGN for classification was performed by removing a column 12, which contains phase shift information (not applicable as no phase plate was used) and adding an absolute magnification value. Reverse conversion from FREALIGN to FREALIGNX for refinement was performed automatically by FREALIGNX. The 4x-binned image stack (62,716 particles) was initially aligned to a ribosome reference (PDB 5U9F)[68] using 5 cycles of mode 3 (global search) alignment including data in the resolution range from 300 to 30 Å until the convergence of the average score. Subsequently, the 4x-binned stack was aligned against the common reference resulting from the previous step, using mode 1 (refine) in the resolution range 300–18 Å (3 cycles of mode 1). In the following steps, the 4x-binned stack was replaced by the 2×-binned image stack, which was successively aligned against the common reference using mode 1 (refine), including gradually increasing resolution limits (5 cycles per each resolution limit; 18-12-10-8 Å) up to 8 Å. 3D density reconstruction was obtained using 60% of particles with highest scores. The refined parameters were used for classification of the 2×-binned stack into 8 classes in 50 cycles using the resolution range of 300–8 Å. This classification revealed six high-resolution classes, one low-resolution (junk) class, and one class representing only 50S subunit (Supplementary Fig. 1a). The particles assigned to the high-resolution 70S classes were extracted from the 2×-binned stack (with >50% occupancy and scores >0) using merge_classes.exe (part of the FREALIGN distribution), resulting in a stack containing 41,382 particles. Classification of this stack was performed for 50 cycles using a focused spherical mask between the A and P sites (mask center coordinates: $x = 191.1$ Å, $y = 224.7$ Å, $z = 159.6$ Å and 30 Å radius, as implemented in FREALIGN). This sub-classification into eight classes yielded two high-resolution classes, which contained both tRNAs and EF-G (Structure II and III); and one high-resolution class, which contained 3 tRNAs (Structure I). For the classes of interest (Structure I, 4263 particles; Structure II, 3179 particles; Structure III, 4612 particles), particles with >50% occupancy and scores >0 were extracted from the 2×-binned stack. Refinement to 6 Å resolution using mode 1 (5 cycles) of the respective 1×-binned stack using 95% of particles with highest scores resulted in ~3.4 Å (Structure I), ~3.5 Å (Structure II) and ~3.4 Å (Structure III) maps (Fourier shell correlation (FSC) = 0.143).

For the frameshifting 70S•mRNA(CCC-A)•fMet-tRNA^fMet•Pro-tRNA^Pro (UGG)•EF-G•GDPCP translocation complex, a dataset of 2591 movies containing 164,504 particles was collected and processed the same way as that for the non-frameshifting complex. All movies were used for further analysis after inspection of the averages and the power spectra computed by CTFFIND4 within cisTEM. The stack and particle parameter files were assembled in cisTEM with the binnings of 1×, 2× and 4× (box size of 400 for unbinned stack). Data classification is summarized in Supplementary Fig. 2. FREALIGNX was used for all steps of particle alignment, refinement, and final reconstruction steps, and FREALIGN v9.11 was used for 3D classification steps[67]. The 4x-binned image stack (164,504 particles) was initially aligned to a ribosome reference (PDB 5U9F) using 5 cycles of mode 3 (global search) alignment including data in the resolution range from 300 to 30 Å until the convergence of the average score. Subsequently, the 4×-binned stack was aligned against the common reference resulting from the previous step, using mode 1 (refine) in the resolution range 300–18 Å (3 cycles of mode 1). In the following steps, the 4×-binned stack was replaced by the 2×-binned image stack, which was successively aligned against the common reference using mode 1 (refine), including gradually increasing resolution limits (5 cycles per each resolution limit; 18-12-10-8 Å) up to 8 Å. 3D density reconstruction was obtained using 60% of particles with highest scores. Subsequently, the refined parameters were used for the classification of the 2×-binned stack into 16 classes in 50 cycles using the resolution range of 300–8 Å. This classification revealed 11 high-resolution classes, 3 low-resolution (junk) classes, and 2 classes representing only the 50S subunit (Supplementary Fig. 2a). The particles assigned to the high-resolution 70S classes were extracted from the 2×-binned stack (with >50% occupancy and scores >0) using merge_classes.exe (part of the FREALIGN distribution), resulting in a stack containing 109,094 particles. Classification of this stack was performed for 50 cycles using a focused spherical mask between the A and P sites (mask center coordinates: $x = 189.5$ Å, $y = 225.0$ Å, $z = 158.3$ Å and 30 Å radius, as implemented in FREALIGN). This sub-classification into eight classes yielded one high-resolution class, which contained both tRNAs and EF-G; and one high-resolution class, which contained three tRNAs (Structure I-FS). The map corresponding to an EF-G-bound translocation state had heterogeneous 30S features corresponding to a mixture of two states (with a highly swiveled and less-swiveled head conformations). The particles assigned to the high-resolution class with both tRNAs and EF-G were extracted from the 2×-binned stack (with >50% occupancy and scores >0) using merge_classes.exe (part of the FREALIGN distribution), resulting in a stack containing 15,088 particles. Classification of this stack was performed for 50 cycles using a 3D mask designed around the head of 30S subunit. This sub-classification into two classes yielded two high-resolution classes, which contained both tRNAs and EF-G but differed in 30S head rotation (Structure II-FS and III-FS). Using subsequent sub-classification of each class into more classes did not yield additional structures. For the classes of interest (Structure I-FS, 12,108 particles; Structure II-FS, 9,059 particles; Structure III-FS, 6,029 particles), particles with >50% occupancy and scores >0 were extracted from

the 2×-binned stack. Refinement to 6 Å resolution using mode 1 (5 cycles) of the respective 1x binned stack using 95% of particles with highest scores resulted in ~3.2 Å (Structure I-FS), ~3.2 Å (Structure II-FS), and ~3.3 Å (Structure III-FS) maps (FSC = 0.143).

In both Structures I and I-FS, E-tRNA density is weak, indicating partial E-site occupancy. This is similar to our previous observations[41], where additional classification resulted in maps with the vacant and tRNA-bound E-site, however, no other differences (i.e., in the occupancy of other sites, or ribosome conformations) were observed. To account for partial density, we have modeled E-site tRNA based on a previous study[41].

For the rotated pre-translocation frameshifting 70S•mRNA(CCA-A)•fMet-tRNA^fMet•Pro-tRNA^Pro(UGG) complex, formed without EF-G, a dataset of 1909 movies containing 178,117 particles was collected on a Titan Krios (FEI) microscope (operating at 300 kV) equipped with the K3 camera system (Gatan), with −0.8 to −2.0 μm defocus. Each exposure was acquired with continuous frame streaming at 25 frames, yielding a total dose of 40.2 e⁻/Å². The nominal magnification was 105,000 and the calibrated super-resolution pixel size at the specimen level was 0.415 Å. The dataset was otherwise collected and processed the same way as that for non-frameshifting or frameshifting complex with EF-G•GDPCP. All movies were used for further analysis after inspection of the averages and the power spectra computed by CTFFIND4 within cisTEM. The stack and particle parameter files were assembled in cisTEM with the binnings of 1×, 2× and 4× (box size of 512 for a unbinned stack). Data classification is summarized in Supplementary Fig. 3. FREALIGNX was used for all steps of particle alignment, refinement and final reconstruction steps and FREALIGN v9.11 was used for 3D classification steps[67]. The 4x-binned image stack (178,117 particles) was initially aligned to a ribosome reference (PDB 5U9F) using 5 cycles of mode 3 (global search) alignment including data in the resolution range from 300 to 30 Å until the convergence of the average score. Subsequently, the 4×-binned stack was aligned against the common reference resulting from the previous step, using mode 1 (refine) in the resolution range 300-18 Å (3 cycles of mode 1). In the following steps, the 4x-binned image stack was replaced by the 2×-binned image stack, which was successively aligned against the common reference using mode 1 (refine), including gradually increasing resolution limits (5 cycles per each resolution limit; 18-12-10-8 Å) up to 8 Å. 3D density reconstruction was obtained using 60% of particles with highest scores. The refined parameters were used for the classification of the 2×-binned stack into 16 classes in 50 cycles using the resolution range of 300-8 Å. This classification revealed 11 high-resolution classes, two low-resolution (junk) classes, and three classes representing only 50S subunit (Supplementary Fig. 3a). The particles assigned to the high-resolution 70S classes were extracted from the 2×-binned stack (with >50% occupancy and scores >0) using merge_classes.exe (part of the FREALIGN distribution), resulting in a stack containing 118,602 particles. Classification of this stack was performed for 50 cycles using a focused spherical mask between the A and P sites (mask center coordinates: $x = 187.0$ Å, $y = 224.9$ Å, $z = 176.9$ Å and 30 Å radius, as implemented in FREALIGN). This sub-classification into eight classes yielded one high-resolution class, which contained a rotated ribosome with one tRNA. The particles assigned to the high-resolution class were extracted from the 2×-binned stack (with >50% occupancy and scores >0) using merge_classes.exe (part of the FREALIGN distribution), resulting in a stack containing 25,345 particles. Classification of this stack was performed for 50 cycles using a focused spherical mask in the P site of the ribosome (mask center coordinates: $x = 179.0$ Å, $y = 225.5$ Å, $z = 178.1$ Å and 40 Å radius). This sub-classification into five classes yielded one high-resolution class, which contained both tRNAs and ribosome in the rotated state (Structure I^rot-FS). For the class of interest (Structure I^rot-FS, 3,658 particles), particles with >50% occupancy and scores >0 were extracted from the 2×-binned stack. Refinement to 6 Å resolution using mode 1 (5 cycles) of the respective 1×-binned stack using 95% of particles with highest scores resulted in ~3.2 Å (Structure I^rot-FS) map (FSC = 0.143).

The maps (Structure I, II, III, I-FS, I^rot-FS, II-FS, and III-FS) were filtered for structure refinements, by blocres and blocfilt from the Bsoft package[69]. To this end, a mask was created for each map by low-pass filtering the map to 30 Å in Bsoft, then binarizing, expanding by three pixels and applying a three-pixel Gaussian edge in EMAN2[70]. Blocres was run with a box size of 20 pixels for all maps. In each case, the resolution criterion was FSC with cutoff of 0.143. The output of blocres was used to filter maps according to local resolution using blocfilt (Supplementary Fig. 4). A range of B-factor values from −50 to −120 Å² was tested for blocfilt maps to achieve optimal balance between higher/lower-resolution regions. Maps sharpened with the B-factor of −80 Å² in bfactor.exe (part of the FREALIGN distribution) were used for model building and structure refinements. FSC curves were calculated by FREALIGN for even and odd particle half-sets.

**Model building and refinement**. Reported cryo-EM structure of *E. coli* 70S•fMet-tRNA^Met•Phe-tRNA^Phe•EF-Tu•GDPCP complex (PDB 5UYM), excluding EF-Tu and tRNAs, was used as a starting model for structure refinement. The structure of EF-G from PDB 4V7D was used as a starting model, and switch regions were generated by homology modeling from PDB 4V9P. The structure of tRNA^Pro (UGG) was created by homology modeling (according to tRNA^Pro (UGG) sequence) using ribosome-bound tRNA^Pro (CGG) (PDB 6ENJ).

Initial protein and ribosome domain fitting into cryo-EM maps was performed using Chimera[71], followed by manual modeling using PyMOL. The linkers between

the domains and parts of the domains that were not well defined in the cryo-EM maps (e.g., loops of EF-G) were not modeled.

All structures were refined by real-space simulated-annealing refinement using atomic electron scattering factors in RSRef[72]. Secondary-structure restraints, comprising hydrogen-bonding restraints for ribosomal proteins and base-pairing restraints for RNA molecules, were employed. Refinement parameters, such as the relative weighting of stereochemical restraints and experimental energy term, were optimized to produce the stereochemically optimal models that closely agree with the corresponding maps. In the final stage, the structures were refined using phenix.real_space_refine[73], followed by a round of refinement in RSRef applying harmonic restraints to preserve protein backbone geometry. Real-space R-factor (RSRef refinement) and correlation coefficient (model-to-map fit—Phenix refinement) were closely monitored to prevent the overfitting of the models to the corresponding maps. The refined structural models closely agree with the corresponding maps, as indicated by low real-space R-factors and high correlation coefficients (Supplementary Table 2). FSC between the final models and maps, and cross-validation half-map FSCs were calculated using Phenix[73], demonstrating good agreement between the structural models and maps (Supplementary Fig. 4). The resulting models have good stereochemical parameters, including low deviation from ideal bond lengths and angles, low number of macromolecular backbone outliers etc., as shown in Supplementary Table 2. Structure quality was validated using MolProbity[74].

Structure superpositions and distance calculations were performed in PyMOL. To calculate the degree of the 30S body rotation or head rotation (swivel) between two 70S structures, the 23S rRNAs or 16S rRNAs of the 30S body were aligned using PyMOL, and the angle was measured in Chimera. These degrees of rotation (30S body/subunit rotation and 30S head rotation) for Structures II, III, I$^{rot}$-FS, II-FS, and III-FS are reported relative to the classical non-rotated Structures I and I-FS, respectively. Figures were prepared in PyMOL and Chimera.

**Reporting summary**. Further information on research design is available in the Nature Research Reporting Summary linked to this article.

## Data availability

The EM density maps generated in this study have been deposited in the EMDB under accession codes EMD-22669 (Structure I); EMD-22670 (Structure II); EMD-22671 (Structure III); EMD-22672 (Structure I-FS); EMD-23528 (Structure I$^{rot}$-FS); EMD-22673 (Structure II-FS); EMD-22674 (Structure III-FS). The atomic coordinates generated in this study have been deposited in the PDB under the accession codes 7K50 (Structure I); 7K51 (Structure II); 7K52 (Structure III); 7K53 (Structure I-FS); 7LV0 (Structure I$^{rot}$-FS); 7K54 (Structure II-FS); 7K55 (Structure III-FS). Source data are provided with this paper.

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

## Acknowledgements

We thank Chen Xu and Kangkang Song for grid screening and data collection at the cryo-EM facility at UMass Medical School; and members of the Korostelev laboratory for discussions and comments on the manuscript. This study was supported and funded by Czech Science Foundation, project no. GJ20-16013Y (to G.D.), and by NIH Grants R35 GM134931 (to Y.M.H.) and R35 GM127094 (to A.A.K.).

## Author contributions

Conceptualization: Y.M.H and A.A.K. Methodology: G.D., H.B.G., A.B.L., I.M., Y.M.H., and A.A.K. Validation: G.D., H.G., Y.M.H., and A.A.K. Investigation: G.D., H. G., A.B.L., I.M., C.E.C., and E.S. Resources: Y.M.H. and A.A.K. Writing- Original Draft: G.D., Y.M. H., and A.A.K. Writing- Review and Editing: All; Visualization: G.D., H.G. Supervision: A.A.K. Funding acquisition: G.D., Y.M.H., and A.A.K.

## Competing interests

The authors declare no competing interests.
