## [Peer Review File · Nature Communications]

REVIEWER COMMENTS

Reviewer #1 (Remarks to the Author):

The manuscript by Demo et al provides structural basis for +1 ribosomal frameshifting during EF-G catalyzed translocation. Authors used AUGCCA-A and AUGCCC-A motifs in mRNA to set up non-frameshifting and frameshifting ribosome complexes, respectively. The structures were prepared using fMet-tRNA, E. coli proline tRNA (UGG) and addition of EF-G with non-hydrolyzable GTP analog GDPCP. The conclusion of the authors is that the ribosome is pre-disposed for +1FS before translocation and that frameshifting is accomplished by the mid-translocation stage of EF-G-catalyzed translocation. The manuscript is written clearly and conclusions are drawn from the presented data, however both results and discussion lack explanations for the previous observation of role of tRNA modifications in +1FS as well as for generality of this mechanism.

1. While the G530 is retracted from the ON position by 2A in the structure I-FS it is not clear is this due to the U34-C3 wobble pair (or some other features of mRNA sequence or structure in vitro or in vivo). How this model with displaced G530 applies to CCC-C sequences and cognate proline tRNA (GGG)? This is not clear in the discussion.
2. Authors could discuss more on the role of tRNA modifications in the proline tRNAs (UGG,GGG,CGG) and how those modifications would contribute to proposed mechanism.
3. On minor note, Figure 4 is not referenced or discussed in the text.

Reviewer #2 (Remarks to the Author):

Demo et al present in the current manuscript cryoEM structures of bacterial ribosomes programmed with mRNAs with and without a frame-shifting prone sequence. A clever experimental design and an optimal use of cryoEM single-particle image processing classification methods allowed the authors to identify 3 structures of the frame-shifting prone mRNA as well as the corresponding 3 structures with the non-frame-shifting prone mRNA. Importantly, these structures were determined using natural tRNAs, including the tRNA^{Pro}(UGG) isoacceptor. The structures reveal very interesting details on how proline codons are decoded and also reveals the mechanism of frameshifting induced along the action of EF-G. The authors were able to visualize how movements of the mRNA take place along tRNA/mRNA translocation catalyzed by EF-G and they were also able to identify bases of the 16S rRNA key in stabilizing a transient "bulged" state of one of the mRNA nucleotides so four nucleotides can be "squeezed" in the E site. Overall, these results explain elegantly the hitherto unknown mechanism used by these sequences to change the reading frame on frameshifting-prone mRNA sequences. The manuscript is succinctly and beautifully written, with a rigorous use of the bibliography. I however think that the authors need to present the cryoEM data as well as the models derived in a more rigorous way. The following suggestion may help in achieving this:

- FSC curves between the final refined model and the final map used for model refinement should be shown ideally alongside with the FSC between half maps.

-A rigorous model overfitting test, ideally re-refining the final model against one half map and comparing with the other half map not included in the refinement, would help in convincing the readers the model is not overfitted, so the conclusion discussed in the manuscript are trustable.

-Final maps colored according to local resolution values should be shown so the reader can judge how homogeneous the maps from the different classes are.

-A supplementary figure with the mask used for classification should be included. This may help in reproducing the classification results by others.

-This reviewer does not understand why the Bfactor value of -80 was used in all maps. A FSC with

randomized phases should be shown so it can be discarded any artefact caused by the mask used in post-processing (<https://pubmed.ncbi.nlm.nih.gov/23872039/>)

-Please, in Table S1 make clear the Map Resolution FSC Criteria is 0.143 and the pixel size 0.525 is the Super-resolution one.

Reviewer #3 (Remarks to the Author):

Frameshifting during translation is an important recoding event. The mechanism of various frameshifting events is only partially understood. In the present paper Demo and colleagues have used cryo-EM to analyze +1-frameshifting on a +1-frameshifting-prone mRNA. They assembled a pre-translocational complex and added EF-G in the presence of GDPCP. Maximum-likelihood classification of the cryo-EM data resulted in one structure of the complex in the pre-translocational state without EF-G and two structures containing EF-G in partially translocated and post-translocated state. Comparison to analogous structures of a non-frameshifting complex reveal that +1 frameshifting occurs during translocation. Overall the results are interesting and add to our understanding of translocation and frameshifting. However, there are a variety of shortcomings that have to be addressed:

Specific Points:

1. There are no functional controls. The authors have to measure the frameshifting efficiency under their experimental conditions. Related to this the question is what is the fraction of complexes observed by cryo-EM that are undergoing frameshift. The measured frame-shifting efficiency has to be discussed in the context of the structural analysis.
2. The authors have used 20 mM MgCl₂ and polyamines to assemble their complexes. What is the rationale to use such non-physiologic buffer condition? What is the impact of the buffer condition on activity (speed and accuracy)? How the authors can be sure that structures obtained under such non-physiologic conditions inform on the mechanism of frameshift under physiologic conditions?
3. A global description of the observed pre-translocational states is missing in the text. According to the figures they are in the non-rotated conformation and contain a deacylated tRNA in the E-site. This is curious as no deacylated tRNA has been used for complex assembly. This should be discussed.
4. As the authors write, the pre-translocational complex undergoes spontaneous rotation of the subunits leading to tRNA hybrid states. However, no such complex is observed. Why? Because rotated complexes have interacted with EF-G? It is thus possible that the observed pre-translocational complexes may have been trapped artificially by contaminating deacylated tRNA in the non-rotated state. To rule out this possibility and to determine the structure of the pre-translocational complex in the rotated state, cryo-EM structures of the pre-translocational complexes without EF-G addition have to be determined.
5. EF-G was stalled on the ribosome using the non-hydrolysable GTP analogue, yet the structure are present in partially or fully translocated states as observed before after GTP hydrolysis in the presence of fusidic acid. How does this relate to current models of translocation according to which GTP hydrolysis precedes tRNA translocation? A discussion should also include a proper description of the present EF-G structures and interactions with the ribosome, the state of the G domain and the structure of the switch regions.
6. Loop1 and 2 of EF-G is not defined.
7. Maps displaying the local resolution of the cryo-EM maps, especially in the region of EF-G and the tRNAs have to be provided.
8. FSC curves between the cryo-EM maps and maps derived from the fitted atomic coordinates have to be provided.
9. From the description of model building and refinement it is not becoming clear how the authors

monitored and prevented overfitting of the models to the corresponding maps.

We thank the reviewers for the thorough review of our manuscript, which helped us improve this work. We have included additional biochemical and cryo-EM data, and revised the manuscript in response to reviewers' criticisms and suggestions. Below, we respond to the specific issues raised by the reviewers.

Reviewer #1 (Remarks to the Author):

The manuscript by Demo et al. provides structural basis for +1 ribosomal frameshifting during EF-G catalyzed translocation. Authors used AUGCCA-A and AUGCCC-A motifs in mRNA to set up non-frameshifting and frameshifting ribosome complexes, respectively. The structures were prepared using fMet-tRNA, E. coli proline tRNA (UGG) and addition of EF-G with non-hydrolyzable GTP analog GDPCP. The conclusion of the authors is that the ribosome is pre-disposed for +1FS before translocation and that frameshifting is accomplished by the mid-translocation stage of EF-G-catalyzed translocation. The manuscript is written clearly and conclusions are drawn from the presented data, however both results and discussion lack explanations for the previous observation of role of tRNA modifications in +1FS as well as for generality of this mechanism.

RESPONSE: Thank you for this generally positive summary. We have added new experiments and expanded our discussion to include tRNA modifications and other issues raised by the reviewer.

1. While the G530 is retracted from the ON position by 2A in the structure I-FS it is not clear is this due to the U34-C3 wobble pair (or some other features of mRNA sequence or structure in vitro or in vivo). How this model with displaced G530 applies to CCC-C sequences and cognate proline tRNA (GGG)? This is not clear in the discussion.

RESPONSE: These are important points, which we now expand upon in Results and Discussion.

In response to the first concern: We agree that other structural or sequence features of mRNA could affect translation and frameshifting. In this work, however, we did not vary mRNA sequence or structure except for the third nucleotide of the mRNA A-site codon, so the difference in the 30S conformation is most likely caused by this single nucleotide substitution resulting in the U34-C3 wobble-position mismatch in the frameshifting complex.

To address the second question about the CCC-C sequence, we have performed experiments to measure the efficiency of frameshifting on different sequences in the cellular and in vitro contexts. Our cell-based experiments show that frameshifting at the CCC-C sequence is negligible when both tRNA^{Pro}(UGG) (encoded by *ProM*) and tRNA^{Pro}(GGG) (encoded by *ProL*) are present, and that it is slightly increased when tRNA^{Pro}(UGG) reads this sequence in the absence of tRNA^{Pro}(GGG). This is consistent with the idea that the complementarity of tRNA^{Pro}(GGG) and the CCC codon leads to efficient stabilization of the codon-anticodon helix by the 30S subunit, resulting in reduced frameshifting relative to the mismatched U34-C3 pair.

We have added these experiments in Figure 1, where we expanded our discussion on the frameshifting frequency on all three codon motifs (CCA-A, CCC-A, and CCC-C). Also, see our detailed responses to Reviewer #3.

2. Authors could discuss more on the role of tRNA modifications in the proline tRNAs (UGG,GGG,CGG) and how those modifications would contribute to proposed mechanism.

RESPONSE: In response to this suggestion, we have added biochemical data testing the role of tRNA modifications and roles of isoacceptor proline tRNAs. *E. coli* expresses three tRNA^{Pro} genes: *ProM* codes for tRNA^{Pro}(UGG), *ProL* codes for tRNA^{Pro}(GGG), and *ProK* codes for tRNA^{Pro}(CGG). As shown in Figure 1c, *ProM* is prone to +1FS at the CCC-A motif, due to the formation of the unstable cmo⁵U34-C3 wobble pairing, and frameshifting is further increased upon loss of m¹G37. In contrast, *ProL* is cognate to the CCC codon, and is thus less prone to +1FS at the CCC-A motif, but more prone to +FS at the CCC-C motif, the frequency of which increases upon loss of m¹G37, as we showed previously (Gamper et al., 2015). *ProK* is not cognate to the CCC codon and does not have a post-transcriptional modification at the wobble C.

The tRNA^{Pro}(UGG) we studied here is encoded by the *ProM* gene and is in the native-state, isolated from *E. coli* cells with the full complement of all of the natural post-transcriptional modifications. Two post-transcriptional modifications that are most relevant to this study are the cmo⁵U34 modification at the wobble position and the methylated guanosine m¹G37 on the 3'-side of the anticodon. Here we show in a cell-based *lacZ* reporter assay (Figure 1) the propensity of +1FS of tRNA^{Pro}(UGG) in the native-state with or without the presence of m¹G37, which in our previous study played an important role in regulating the propensity of +1FS (Gamper et al., 2015). In this *lacZ* reporter construct, a CCX-X codon motif was inserted next to the AUG start codon, such that a +1FS event was necessary to fully translate the *lacZ* gene to synthesize β-galactosidase (β-gal), whereas lack of +1FS would lead to premature termination of protein synthesis (Figure 1a).

In *E. coli* cells expressing the CCA-A reporter, the co-existence of *ProM* for tRNA^{Pro}(UGG), which would read the CCA codon, and *ProL* for tRNA^{Pro}(GGG), which would read the CCC codon, showed a background level of +1FS, which did not change much when *ProL* was deleted. This background level of +1FS remained stable even upon deletion of m¹G37 (Figure 1b), indicating that *ProM* alone was sufficient to read the CCA codon, which is cognate to its anticodon. In *E. coli* cells expressing the CCC-A reporter, where the CCC codon is cognate to *ProL* and can be read by *ProM* through the cmo⁵U34-C3 pairing, the absence of *ProL* increased +1FS and this increase was enhanced by the absence of m¹G37 (Figure 1c). This supports the notion that translation of the CCC codon by *ProM* alone is prone to +1FS and consistent with the notion that the propensity of +1FS by *ProM* is increased upon loss of m¹G37. In *E. coli* cells expressing the CCC-C reporter, the absence of *ProL* showed only a small increase in +1FS, indicating that *ProL* is the primary reader of the CCC codon. Loss of m¹G37 increased +1FS to a similar level whether *ProL* was present or not (Figure 1d), supporting the notion that *ProM* was responsible for reading the CCC codon in the absence of *ProL* and that m¹G37 is critical for reading-frame maintenance of *ProM*.

3. On minor note, Figure 4 is not referenced or discussed in the text.

RESPONSE: We thank the reviewer for the comment, we have added call-outs to Figure 6 (Figure 4 in the initial manuscript) in the last section of our discussion.

Reviewer #2 (Remarks to the Author):

Demo et al present in the current manuscript cryoEM structures of bacterial ribosomes programmed with mRNAs with and without a frame-shifting prone sequence. A clever experimental design and an optimal use of cryoEM single-particle image processing classification methods allowed the authors to identify 3 structures of the frame-shifting prone mRNA as well as the corresponding 3 structures with the non-frame-shifting prone mRNA. Importantly, these structures were determined using natural tRNAs, including the tRNA^{Pro}(UGG) isoacceptor. The structures reveal very interesting details on how proline codons are decoded and also reveals the mechanism of frameshifting induced along the action of EF-G. The authors were able to visualize how movements of the mRNA take place along tRNA/mRNA translocation catalyzed by EF-G and they were also able to identify bases of the 16S rRNA key in stabilizing a transient "bulged" state of one of the mRNA nucleotides so four nucleotides can be "squeezed" in the E site. Overall, these results explain elegantly the hitherto unknown mechanism used by these sequences to change the reading frame on frameshifting-prone mRNA sequences.

The manuscript is succinctly and beautifully written, with a rigorous use of the bibliography.

I however think that the authors need to present the cryoEM data as well as the models derived in a more rigorous way. The following suggestion may help in achieving this:

1. FSC curves between the final refined model and the final map used for model refinement should be shown ideally alongside with the FSC between half maps.

RESPONSE: We thank the reviewer for this suggestion. We have added FSC curves between final refined models and the final map to Supplementary figure 4. We show the maps alongside the FSC between half maps.

2. A rigorous model overfitting test, ideally re-refining the final model against one half map and comparing with the other half map not included in the refinement, would help in convincing the readers the model is not overfitted, so the conclusion discussed in the manuscript are trustable.

RESPONSE: Thank you. We have re-refined the models and show the FSC curves for half maps (self and cross-validation) in Supplementary figure 4 alongside the FSC between half maps and FSC of the map-to-model fit. We have accordingly expanded the Methods section "Model building and refinement".

3. Final maps colored according to local resolution values should be shown so the reader can judge how homogeneous the maps from the different classes are.

RESPONSE: Final maps colored according to local resolution values are now shown in Supplementary figure 4 together with an example of local resolution in large ribosomal subunit.

4. A supplementary figure with the mask used for classification should be included. This may help in reproducing the classification results by others.

RESPONSE: We thank the reviewer for the comment. We now show the used focus masks and their parameters in each classification scheme (Supplementary figures 1, 2 and 3). We also specify the mask center coordinates in Methods section "Cryo-EM and image processing".

5. This reviewer does not understand why the B-factor value of -80 was used in all maps. A FSC with randomized phases should be shown so it can be discarded any artefact caused by the mask used in post-processing.

RESPONSE: We thank the reviewer for the comment. We now explain in Methods that we have generated maps with different B-factors from -50 to -120 and found that applying the B-factor of -80 results in most detailed maps and results in better agreement with atomic models.

As for the post-processing mask artefact concern, we note that we used cisTEM/Frealign for data processing, where FSC calculation is performed differently from Relion. Instead of using a tight mask (as in Relion) to get the best possible FSC, cisTEM/Frealign applies a generous spherical mask to calculate the "masked" FSC on the volume of particle inside the mask, as described in refs: Grant et al., eLife 2018, and Grigorieff, Methods in Enzymology, 2016. As previously described, masking artifacts are completely avoided and the randomization test is unnecessary. Accordingly, CisTEM/Frealign FSC calculation was considered a more reliable method and recommended by the EMBL cryo-EM meeting "Frontiers in cryoEM Validation" in January 2019 in the UK.

6. Please, in Table S1 make clear the Map Resolution FSC Criteria is 0.143 and the pixel size 0.525 is the Super-resolution one.

RESPONSE: We have added to Supplementary Table 1 the map resolution FSC criterion and the super resolution pixel size.

Reviewer #3 (Remarks to the Author):

Frameshifting during translation is an important recoding event. The mechanism of various frameshifting events is only partially understood. In the present paper Demo and colleagues have used cryo-EM to analyze +1-frameshifting on a +1-frameshifting-prone mRNA. They assembled a pre-translocational complex and added EF-G in the presence of GDPCP. Maximum-likelihood classification of the cryo-EM data resulted in one structure of the complex in the pre-translocational state without EF-G and two structures containing EF-G in partially translocated and post-translocated state. Comparison to analogous structures of a non-frameshifting complex reveal that +1FS occurs during translocation. Overall the results are interesting and add to our understanding of translocation and frameshifting. However, there are a variety of shortcomings that have to be addressed:

Specific Points:

1. There are no functional controls. The authors have to measure the frameshifting efficiency under their experimental conditions. Related to this the question is what is the fraction of complexes observed by cryo-EM that are undergoing frameshift. The measured frame-shifting efficiency has to be discussed in the context of the structural analysis.

RESPONSE: We thank the reviewer for raising this important point. We have performed cellular assays (Figure 1) and biochemical assays (Figure 2), which we now describe in the manuscript. Concerning the cellular assays, please see our response to Reviewer 1 above.

As for biochemical assays, we used an *E. coli* in vitro translation system composed of purified components and supplemented it with requisite tRNAs and translation factors to perform kinetic experiments. We delivered EF-G*GTP and an equimolar mixture of three ternary complexes (TCs), each consisting of EF-Tu-GTP with Pro-tRNA^{Pro}(UGG), Ser-tRNA^{Ser}, or Val-tRNA^{Val}, to a 70S ribosome initiation complex (70SIC) that placed the initiator fMet-tRNA^{fMet} in the P site (Figure 2a). The 2nd codon in the mRNA was either CCC or CCA for Pro, and the 3rd codon was AGU for Ser. After peptidyl transfer and formation of a pre-translocation complex that placed fMP-tRNA^{Pro}(UGG) in the A site, EF-G-GTP would catalyze translocation to move fMP-tRNA^{Pro}(UGG) to the P site. As soon as the post-translocation complex was formed, Ser- and Val-TC would compete for the codon at the A site to promote formation of an fMPS or an fMPV tripeptide. Depending on the reading-frame of fMP-tRNA^{Pro}(UGG) in the post-complex, the synthesis of fMPS would report on the sub-population of tRNA^{Pro}(UGG) that remained in the 0-frame, whereas the synthesis of fMPV would report on the sub-population that shifted to the +1-frame. We calculated the fractional conversion of fMP to fMPV as the % of +1FS.

The results showed +1FS at the CCC-A codon motif at both the high-fidelity buffer (HF, 3.5 mM Mg²⁺) and cryo-EM buffer (CE, 20 mM Mg²⁺) at 37 °C, the latter of which were the conditions where the cryo-EM complexes were made. We showed that +1FS was decreased at 20 °C (Figure 2b, c), indicating that the higher thermal motion of the ribosome at 37 °C facilitated +1FS. In contrast, there was no evidence of +1FS at the CCA-A codon motif in either buffer (Figure 2d), supporting the notion that the fully complementary pairing of the native-state tRNA^{Pro}(UGG) is stable and does not promote +1FS. The measured frequency of +1FS at the CCC-A motif in the cryo-EM condition in the CE buffer at 37 °C was 5%, during active translocation reaction in the presence of EF-G-GTP. By contrast, translocation was stalled in the cryo-EM work by GDPCP, resulting in ribosomes sampling predominantly the frameshifted state. Together, these findings suggest that the inhibition of EF-G dissociation by GDPCP enhances +1FS, whereas rapid and transient EF-G binding/dissociation is important for the maintenance of the mRNA frame.

We also performed experiments to evaluate the importance of post-transcriptional modifications in tRNA^{Pro}(UGG). The tRNA used in the structural analysis was in the native-state, containing the full complement of all of natural post-transcriptional modifications. For this evaluation, we determined the frequency of +1FS of tRNA^{Pro}(UGG) in the transcript-state, which was made by in vitro transcription lacking any post-transcriptional modification. We showed that the tRNA^{Pro}(UGG) transcript exhibited an increased yield of +1FS relative to the native-state at the CCC-A motif in the HF buffer (Figure 2e vs. 2c), supporting the notion that post-transcriptional modifications are generally important to suppress +1FS. This notion is further supported by our kinetic assay for +1FS, showing that loss of post-transcriptional modifications in the transcript-

state enhanced the rate of +1FS (Supplementary Figure 6). Notably, the tRNA^{Pro}(UGG) transcript was unable to produce stoichiometric synthesis of the in-frame fMPS on the CCA-A motif, (Supplementary Figure 6), emphasizing the importance of modifications in translation. These results demonstrate the value of studying the native-state tRNA^{Pro}(UGG) in our structural analysis, where natural post-transcriptional modifications support stoichiometric synthesis of the in-frame fMPS (Supplementary Figure 6, while also enabling a detectable level of +1FS in cryo-EM conditions (Supplementary Figure 6).

2. The authors have used 20 mM MgCl₂ and polyamines to assemble their complexes. What is the rationale to use such non-physiologic buffer condition? What is the impact of the buffer condition on activity (speed and accuracy)? How can the authors be sure that structures obtained under such non-physiologic conditions inform on the mechanism of frameshift under physiologic conditions?

RESPONSE: We used the standard conditions that are commonly used in structural studies of the ribosome to capture structural intermediates. Although buffer conditions are indeed likely to shift the equilibrium between the populations of functional states, the structures of these functional states usually agree with biochemical studies.

Nevertheless, to address this reviewer's point, we have performed a series of biochemical experiments to compare frameshifting under near-physiological conditions (i.e. 3.5 mM Mg) and cryo-EM conditions. As we describe in our response to comment #1, +1FS was observed both in the high-fidelity buffer (HF, 3.5 mM Mg²⁺) and cryo-EM buffer (CE, 20 mM Mg²⁺) at 37 °C, i.e. the conditions under which the cryo-EM complexes were made. Yet, as expected, increasing Mg²⁺ concentration from 3.5 mM to 20 mM progressively decreased the yield of fMPS synthesis in both the HF buffer and the CE buffer, supporting the notion that high Mg²⁺ concentrations inhibit ribosome conformational movements that are required for protein synthesis. Although the frequency of +1FS at the CCC-A motif in the cryo-EM condition in the CE buffer at 37 °C was 5%, it was measured in an active translocation reaction. By contrast, translocation was stalled in our cryo-EM complexes by the presence of GDPCP, which inhibits EF-G from dissociation, enabling us to capture intermediates of translocation.

3. A global description of the observed pre-translocational states is missing in the text. According to the figures they are in the non-rotated conformation and contain a deacylated tRNA in the E-site. This is curious as no deacylated tRNA has been used for complex assembly. This should be discussed.

RESPONSE: We have expanded the discussion of pre-translocation states, and included a newly determined *rotated* pre-translocation structure (see our response to the next comment). We also discussed the presence of sub-stoichiometric deacylated tRNA in the E-site in Methods. We note that partial occupancy of the E site by non-cognate deacylated tRNA to the E site is common in structural studies.

4. As the authors write, the pre-translocational complex undergoes spontaneous rotation of the subunits leading to tRNA hybrid states. However, no such complex is observed. Why? Because

rotated complexes have interacted with EF-G? It is thus possible that the observed pre-translocational complexes may have been trapped artificially by contaminating deacylated tRNA in the non-rotated state. To rule out this possibility and to determine the structure of the pre-translocational complex in the rotated state, cryo-EM structures of the pre-translocational complexes without EF-G addition have to be determined.

RESPONSE: We thank the reviewer for raising this important point. In our pre-translocation complexes formed with EF-G*GDPCP, there are no detectable rotated ribosomes because they have been depleted by binding of EF-G*GDPCP, as expected.

To capture the rotated state in a pre-translocation complex, we have performed cryo-EM studies with the frameshifting mRNA *without EF-G*. As expected, classification has revealed a mixture of rotated and non-rotated pre-translocation ribosomes. We have determined a cryo-EM structure of the rotated pre-translocation state (Structure I^{rot}-FS), which reveals that the conformation of the 30S decoding center is similar to that in the non-rotated pre-translocation state (I-FS), where the anticodon-codon helix is in the 0-frame and the 30S is in the open conformation. This structure therefore supports the conclusion that there is no frameshifting at the A site, despite the open conformation of the 30S. The new structure is now described in section “Pre-translocation frameshifting structures adopt an open 30S conformation”.

5. EF-G was stalled on the ribosome using the non-hydrolysable GTP analogue, yet the structure are present in partially or fully translocated states as observed before after GTP hydrolysis in the presence of fusidic acid. How does this relate to current models of translocation according to which GTP hydrolysis precedes tRNA translocation? A discussion should also include a proper description of the present EF-G structures and interactions with the ribosome, the state of the G domain and the structure of the switch regions.

RESPONSE: We extended the section “mRNA frame is shifted in the EF-G-bound structures II-FS and III-FS” to indicate that our structures resemble previous structures in which EF-G is locked by an antibiotic (e.g. fusidic acid, or neomycin), non-hydrolyzable analog (e.g. GDPNP), or catalytically dead EF-G mutant (H92A). Because GDPCP cannot be hydrolyzed, the structures do not report on the timing of GTP hydrolysis during translocation. We also describe the conformation of the GTPase center and provide additional density figures in Supplementary Figure 9.

6. Loop1 and 2 of EF-G is not defined.

RESPONSE: Loops 1 and 2 of EF-G are now defined in the main text of the manuscript specifically in section “mRNA frame is shifted in the EF-G-bound structures II-FS and III-FS” and also in Figure 4.

7. Maps displaying the local resolution of the cryo-EM maps, especially in the region of EF-G and the tRNAs have to be provided.

RESPONSE: Cryo-EM maps colored according to local resolution values are now shown in Supplementary figure 4 with applied slab cuts to specifically show the local resolutions for EF-G and tRNAs. We also show local density for EF-G GTPase in Supplementary Figure 9.

8. FSC curves between the cryo-EM maps and maps derived from the fitted atomic coordinates have to be provided.

RESPONSE: The FSC curves between final refined model and the final map; FSC curves for half maps (self and cross-validation) are now shown in Supplementary figure 4 alongside the FSC between half maps.

9. From the description of model building and refinement it is not becoming clear how the authors monitored and prevented overfitting of the models to the corresponding maps.

RESPONSE: We have expanded Methods and include an additional Supplementary Figure, showing FSCs between models refined against half maps, with the corresponding maps and maps close-up views.

Reference:

Gamper, H. B., Masuda, I., Frenkel-Morgenstern, M. & Hou, Y. M. Maintenance of protein synthesis reading frame by EF-P and m(1)G37-tRNA. *Nat Commun* **6**, 7226, doi:10.1038/ncomms8226 (2015).

REVIEWERS' COMMENTS

Reviewer #1 (Remarks to the Author):

Authors have fully answered my concerns. Additional experiments and discussion strengthened the original manuscript.

Reviewer #3 (Remarks to the Author):

The authors are to be commended for their careful and thorough revisions. The addition of new functional and structural data has benefited the paper greatly. The paper is now ready for publication.